# Envelope-specific B-cell populations in African green monkeys chronically infected with simian immunodeficiency virus

Ruijun Zhang[1], David R. Martinez[1], Quang N. Nguyen[1], Justin Pollara[1,2], Trina Arifin[1], Christina Stolarchuk[1], Andrew Foulger[1], Josh D. Amos[1], Robert Parks[1], Jonathon E. Himes[1], Minyue Wang[1], Regina W. Edwards[1], Ashley M. Trama[1], Nathan Vandergrift[1], Lisa Colvin[1], Ken Dewar[3,4,5], Nikoleta Juretic[3,4,5], Jessica Wasserscheid[3,4,5], Guido Ferrari[1,2], Hua-Xin Liao[1,*] & Sallie R. Permar[1,6,*]

African green monkeys (AGMs) are natural primate hosts of simian immunodeficiency virus (SIV). Interestingly, features of the envelope-specific antibody responses in SIV-infected AGMs are distinct from that of HIV-infected humans and SIV-infected rhesus monkeys, including gp120-focused responses and rapid development of autologous neutralization. Yet, the lack of genetic tools to evaluate B-cell lineages hinders potential use of this unique non-human primate model for HIV vaccine development. Here we define features of the AGM Ig loci and compare the proportion of Env-specific memory B-cell populations to that of HIV-infected humans and SIV-infected rhesus monkeys. AGMs appear to have a higher proportion of Env-specific memory B cells that are mainly gp120 directed. Furthermore, AGM gp120-specific monoclonal antibodies display robust antibody-dependent cellular cytotoxicity and CD4-dependent virion capture activity. Our results support the use of AGMs to model induction of functional gp120-specific antibodies by HIV vaccine strategies.

[1] Duke Human Vaccine Institute, Duke University School of Medicine, Durham, North Carolina 27710, USA. [2] Department of Surgery, Duke University Medical Center, Durham, North Carolina 27710, USA. [3] Research Institute of the McGill University Health Centre, McGill University and Génome Québec Innovation Centre, Montreal, Québec, Canada H3A 0G1. [4] Department of Human Genetics, McGill University, Montreal, Québec, Canada H3A 1B1. [5] Department of Experimental Medicine, McGill University, Montreal, Québec, Canada H3A 1A3. [6] Department of Pediatrics, Duke University Medical Center, Durham, North Carolina 27705, USA. * These authors jointly supervised this work. Correspondence and requests for materials should be addressed to H.X.L. (email: hliao@duke.edu) or to S.R.P. (email: sallie.permar@duke.edu).

A critical priority for human immunodeficiency virus (HIV) vaccine development is the elicitation of broadly neutralizing antibodies (bnAbs). Although broadly neutralizing serum responses arise in more than half of chronically HIV-infected individuals[1], to date no HIV vaccine concept has successfully elicited bnAbs in human and non-human primate (NHP) trials[2,3]. The majority of the broad neutralizing epitopes have mapped to the gp120 subunit of envelope (Env)[4]. Yet, recent studies have demonstrated that a pre-existing pool of antibody responses against the gut microbiota cross-reacts with the gp41 subunit of the HIV Env complex in both acutely HIV-infected patients[5] and HIV Env vaccine recipients[6]. Thus, more studies are needed to better understand how to elicit HIV Env-specific antibodies against neutralizing gp120 epitopes. Defining the roadmap for how gp120 epitope-specific bnAbs are produced will be important in designing potential strategies to induce broadly reactive HIV antibodies. Here we introduce a unique NHP model and develop new tools to help define the elicitation of gp120-directed antibody responses, an initial step in eliciting gp120-directed neutralizing responses.

African green monkeys (AGMs), a natural primate host of simian immunodeficiency virus (SIV), have co-evolved with the virus for more than 30,000 years, resulting in a number of host adaptions to mitigate disease progression. In contrast to the high propensity of vertical HIV transmission in humans, natural SIV hosts only rarely transmit the virus to their infants, despite the virus being consistently present in plasma and breast milk[7,8]. Unlike non-natural SIV/HIV hosts, which include rhesus monkeys (RMs), SIV-infected AGMs do not display B-cell dysfunction or hypergammaglobulinemia during chronic infection[9]. Interestingly, the initial B-cell responses in AGMs are predominantly directed against the SIV Env gp120 (ref. 10), compared with the initial gp41-focused response in humans and RMs[5]. Moreover, AGMs develop autologous neutralizing responses in plasma and breast milk more rapidly than SIV-infected RMs. Thus, AGMs are a potentially unique NHP model for defining induction pathways of antibody responses to SIV/HIV infection and vaccination. Moreover, interrogating the Env-specific memory B cells in AGMs may also provide insight into virus-specific antibody responses that evolved over time to optimally target SIV and potentially contribute to the containment of disease pathogenesis.

For detailed analysis of B-cell lineage evolution in preclinical vaccine development in NHP models, it is essential to define the immunoglobulin (Ig) germline genes of the NHP species and their relationship to that of humans. The Ig loci of RMs has recently been assembled[11] and a more accurate database of heavy chain variable ($V_H$) germline genes was recently defined[12], making it possible to accurately assess the similarity of vaccine-elicited antibody responses in RMs and humans. Yet, these studies cannot be performed in natural SIV hosts due to the lack of $V_H$ germline gene database. In this study, we identify the constant and variable gene segments of the Ig $V_H$ and light-chain variabl$r$ ($V_L$) in the recently sequenced AGM genome[13] and compare the genetic distribution of Ig genes in this animal species with those of RMs and humans. We then investigate the unbiased memory B-cell populations and compare the proportion of Env-specific B cells across SIV AGMs, RMs and humans. Env-specific monoclonal antibodies in chronically SIV-infected AGMs are further investigated by defining the epitope specificity and antiviral functions of isolated Env-reactive monoclonal antibodies. Our findings reveal that AGMs appear to have a higher proportion of Env-specific memory B cells. Interrogating the nature of this gp120-biased response in natural SIV hosts could assist in the development of vaccination strategies in humans aimed at eliciting functional gp120-specific responses.

## Results

**Definition of AGM immunoglobulin gene loci.** We identified the Ig gene loci in the newly sequenced AGM genome and annotated AGM Ig heavy- and light-chain germline genes and constant regions from the assembly of version 1.0 of the AGM genome[13] by aligning with the published human Ig germline genes (allele 01)[14–16] and constant region sequences. The AGM heavy-chain locus was located at the telomeric extremity of the long arm of chromosome 24 (CHR24) (Supplementary Fig. 1) with respect to the human locus. Out of the six identified heavy-chain constant ($C_H$) segments, five were located on CHR24 and one was from an unplaced contig in an earlier version of the genome assembly (PART 12 of the 4.0.3 Newbler assembly). These six $C_H$ segments were clearly classified as IgA, IgD, IgE, IgG1, IgG2 and IgM (Fig. 1a) when aligned with human and RM $C_H$ segments (Supplementary Fig. 2). However, only a single IgA germline gene was identified in AGMs compared with two IgA subclasses in humans and a single IgA subclass with considerable polymorphism in RMs[17]. Only two IgG subclasses were identified in AGMs compared with four identified IgG subclasses, both in humans and RMs (Fig. 1a). The homology of the $C_H$ sequences of each antibody isotype in AGMs was much closer to that of RMs than humans, including the two IgG subclasses that separately segregated with IgG1 and IgG2 (Fig. 1b,c and Supplementary Table 2).

We identified the AGM core sequences of Eµ (cEµ) located between $J_H$ and $C_H$ loci. Interestingly, AGM cEµ elements were

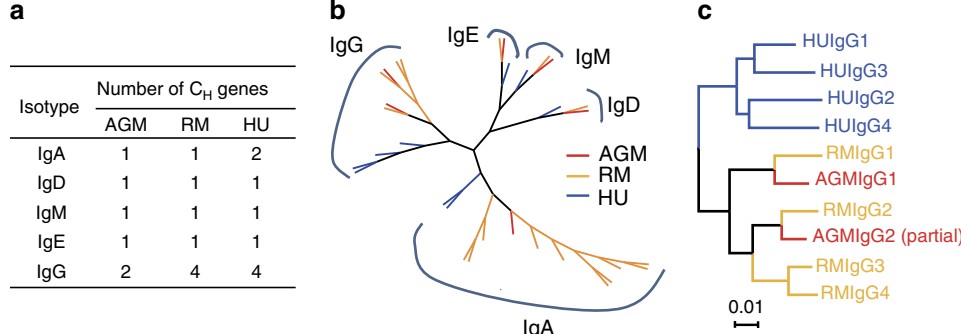

**Figure 1 | Genetic comparison of immunoglobulin heavy chain constant regions ($C_H$) of AGM, RM and human.** (**a**) Number of immunoglobulin $C_H$ isotype genes identified in AGM ($n = 6$), RM ($n = 8$) and human ($n = 9$). (**b**) Joint radial phylogenic tree of AGM (red lines), RM (orange lines) and human (blue lines) $C_H$ genes. Ten RM IgA intra-species heterogenetic alleles were listed in the radial tree. (**c**) Rectangular phylogenic tree of AGM, RM and human IgG subclass $C_H$ genes, demonstrating greater homology of AGM $C_H$ genes to that of RM than human.

| Isotype | Number of $C_H$ genes | | |
|---|---|---|---|
| | AGM | RM | HU |
| IgA | 1 | 1 | 2 |
| IgD | 1 | 1 | 1 |
| IgM | 1 | 1 | 1 |
| IgE | 1 | 1 | 1 |
| IgG | 2 | 4 | 4 |

highly homologous to that of RM and human cEµ elements with only a single nucleotide difference in the µA element between AGM and RM (Supplementary Fig. 3)[18]. The high conservation of cEµs among these three species suggests that this enhancer may have a critical role in Ig recombination in primates.

Functional Ig $V_H$ genes were identified in the AGM genome using human Ig germline gene database adopted from the international ImMunoGeneTics information system (IMGT)[19]. Based on the human Ig database, we identified a total of 58 functional AGM $V_H$ segments belonging to 5 $V_H$ families, but none belonging to $V_H6$ and $V_H7$ germline families (Fig. 2a and

Supplementary Fig. 1). To identify differences in $V_H$ germline genes, we compared the AGM and human Ig $V_H$ gene maps[20] and found a gap in the centromeric part of the $V_H$ gene locus where human functional $V_H6$ and $V_H7$ genes are present (Supplementary Fig. 4). A total of 37 functional diversity ($D_H$) segments and 6 joining ($J_H$) segments were also identified in the region between $V_H$ locus and $V_C$ locus in AGM CHR24 (Supplementary Fig. 1 and Supplementary Data 1). The rectangular structure of the phylogenetic tree of 58 AGM $V_H$ genes closely resembles that of humans and RMs[12] (Fig. 2b). When $V_H$ genes from these three species were displayed in a joint

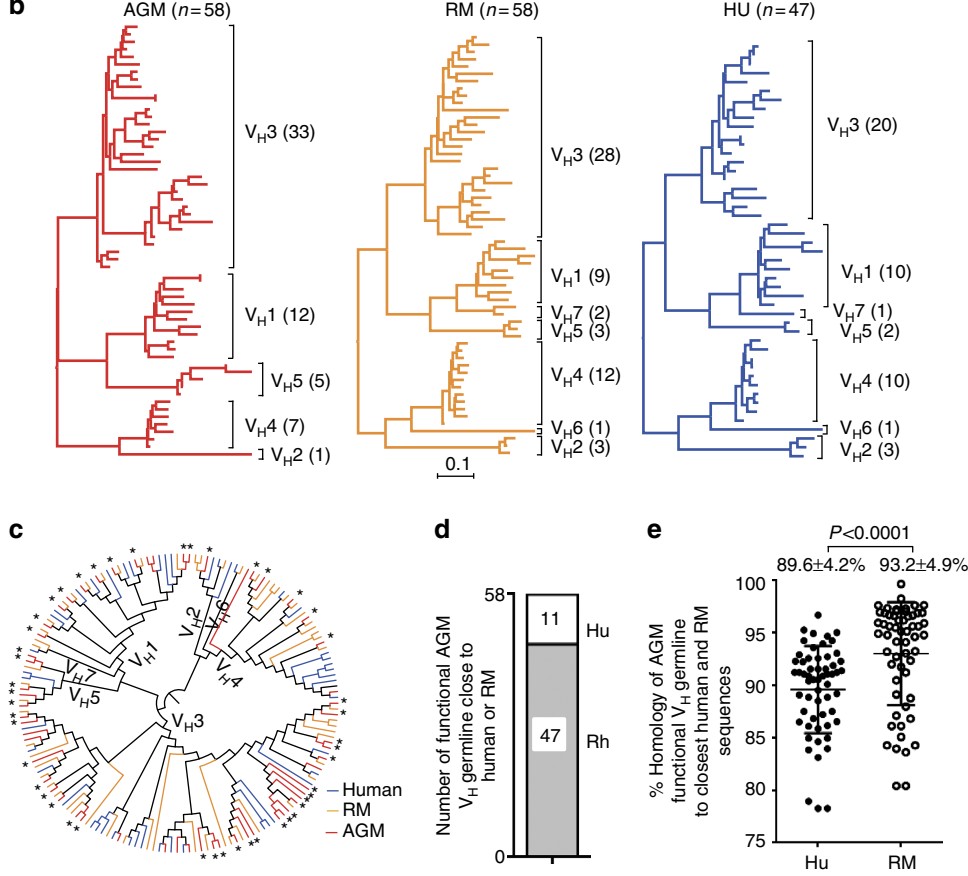

**Figure 2 | Genetic comparison of $V_H$ genes of AGM, RM and human.** (**b**) Number of functional $V_H$ genes in three species. (**b**) Phylogenetic comparison of AGM ($n=47$), RM ($n=58$) and AGM ($n=58$) $V_H$ families. (**c**) Joint radial phylogenic trees of AGM (red lines), RM (orange lines) and human (green lines) $V_H$ genes. The AGM $V_H$ genes that are genetically more similar to RM ($n=47$) than human ($n=11$) $V_H$ homologues are highlighted with stars (*). (**d**) Number of AGM $V_H$ genes that are genetically more similar to human (white box) or RM (grey box) than the other species. (**e**) Percent homology of AGM $V_H$ genes to those of RM (empty circle, $93.2\pm4.9\%$) and human (filled circle, $89.6\pm4.2\%$; $P<0.0001$) were indicated and compared by an exact Wilcoxon rank sum test. The data in **e** shows means ± s.d.

phylogenetic tree, the $V_H$ genes distributed into different branches by gene families rather than by species (Fig. 2c). Overall, a substantially higher number of AGM genes are more homologous to those of RMs (44/58) than humans (14/58) (Fig. 2d) with an average percentage homology between AGM $V_H$ genes and the closest functional RM or human $V_H$ genes of $93.2 \pm 4.6\%$ and $89.6 \pm 4.2\%$, respectively ($P < 0.0001$, Wilcoxon-signed rank test) (Fig. 2e). Finally, we identified 64 functional κ-chain variable (Vκ), 5 functional joining ($J_κ$) and one functional κ-chain constant ($C_κ$) segments in AGM CHR14 (Supplementary Figs 5 and 7, and Supplementary Data 2), as well as 45 functional λ-chain variable ($V_λ$), 6 joining ($J_λ$) and 4 λ-chain constant ($C_λ$) segments in AGM CHR19 (Supplementary Figs 6 and 8, and Supplementary Data 3). Functional AGM $V_κ$ and $V_λ$ genes were also more homologous to those of RMs than humans (Supplementary Figs 5E and 6E). Average percent homology of AGM $V_κ$ to those of RMs and humans was $94.5 \pm 4.5\%$ and $91.1 \pm 5.0\%$, respectively ($P < 0.0001$, Wilcoxon-signed rank test); the average percentage homology of AGM $V_λ$ to those of RMs and humans was $94.6 \pm 6.3\%$ and $90.7 \pm 5.6\%$, respectively ($P < 0.0001$, Wilcoxon-signed rank test).

**High proportion of gp120-specific B cells in AGMs**. We previously showed that SIV-infected AGMs have a predominantly gp120-focused antibody response during acute and chronic SIV infection[9,10]. This is in contrast to the predominantly gp41-specific antibody responses of HIV/SIV-infected human and RMs, which have been associated with ineffective, polyspecific antibody responses following acute infection[5,6]. Furthermore, the autologous virus neutralizing responses appeared by 6 weeks after infection and was more than a log higher in plasma of AGMs compared with RMs by 1 year post infection[9]. Autologous neutralizing responses were detectable in breast milk of chronically infected AGMs, but not in the milk of RMs. Although the antibody function is likely to be affected by the more prominent $CD4^+$ T-cell depletion and B-cell dysfunction in SIV-infected RMs, the distinct specificity of the responses suggests differences in the immunoglobulin repertoires of AGMs, compared with RMs SIV infection[9,10].

To compare the unbiased memory B-cell populations of chronically SIV-infected AGMs with that of SIV/HIV-infected non-natural hosts RMs and humans, AGM $V_H$, Vκ and Vλ genes were amplified from single memory B cells ($CD3^- CD20^+ IgD^- CD27^{all}$) sorted from blood and breast milk of AGMs and RMs 1 year post SIV infection and chronically HIV-1-infected humans (Supplementary Fig. 9). As we found that the $V_H$ and $V_L$ genes of AGM are closer to those of RM than to those of humans, AGM $V_H$ and $V_L$ gene pairs were amplified by reverse transcriptasePCR (RT–PCR) using RM[21] Ig primers as described previously and produced as recombinant IgG using the strategy outlined in Fig. 3a (refs 21,22). Monoclonal antibodies produced from blood and breast milk B cells (Fig. 3b) of SIV-infected AGMs, RMs and HIV-1-infected individuals were screened for binding to the autologous SIV Env gp120 and gp140 (SIVsab92018ivTF or SIVmac251 for AGMs and RMs, respectively) and the autologous linear SIV sp400 peptide (gp41 immunodominant epitope), clade-matched HIV-1 Env gp120, gp140 and MN gp41 proteins, group M consensus envelopes that had broad reactivity with different HIV-1 Env antibodies and HIV-1-positive sera from different HIV-1 subtypes[23], and HIV-1 gp41 immunodominant region peptide sp400. In total, from the sorted single memory B cells, 175 monoclonal antibodies (79 from blood and 96 from breast milk) from 4 SIV-infected AGMs, 109 monoclonal antibodies from blood of 2 SIV-infected RMs and 244 monoclonal antibodies (203 from blood and 41 from breast

milk) from 7 HIV-infected individuals were isolated and screened for SIV/HIV Env reactivity (Fig. 3b and Supplementary Table 1). Remarkably, the proportions of SIV/HIV Env-specific monoclonal antibodies isolated from blood (19.0%, 15/79) and breast milk (30.2%, 29/96) memory B cells of SIV-infected AGMs were considerably higher than those of memory B cells isolated from blood (0.5%, 1/203) and breast milk (7.3%, 3/41) from chronically HIV-infected individuals ($P < 0.0001$ and $P = 0.01$, Fisher's exact test, respectively) (Fig. 3b). Moreover, there were higher proportions of blood and breast milk Env-specific antibodies directed against gp120 (11.4% and 10.4%, respectively) than gp41 (3.8% and 5.2%, respectively) (Fig. 3b) in AGMs, whereas the reactivity to gp41 was the only specificity of Env-reactive antibodies isolated from blood memory B cells in humans (Fig. 3b). In contrast, both gp120 and gp41-specific antibodies were isolated from breast milk memory B cells of HIV-infected lactating women (4.9% and 2.4%, respectively), consistent with a previous report[24]. Interestingly, although the proportion of Env-specific peripheral blood B cells was similar between SIV-infected AGMs (19%) and RMs (19.3%), the proportion of gp120-specific antibodies isolated from AGM blood memory B cells (11.4%) was significantly higher than that of RMs (0.9%, $P = 0.001$, Fisher's exact test) (Fig. 3b). Interestingly, we compared the total number of B cells ($CD3^- CD20^+$) and the total number of memory B cells ($CD20^+ IgD^- CD27^+$) between the chronically SIV-infected AGMs and RMs, and observed no statistical difference (Supplementary Table 6).

**Genetic diversity and mutation of AGM Ig variable genes**. It is well described that HIV-specific antibodies isolated from infected humans that have broadly neutralizing activity are typically IgG, are highly somatically mutated and display long heavy chain complementarity-determining region 3 (HCDR3)[2,25–28]. Thus, to address the genetic characteristics and degree of antibody maturation of the Env-specific monoclonal antibodies isolated from AGMs, the newly identified AGM constant and variable regions of Ig heavy chain, κ-, and λ-light chain were used to annotate the isolated Ig heavy and light chain pairs, and calculate their somatic hypermutation (SHM) frequency in AGMs. We found that IgG was the dominant isotype of the antibodies isolated from both AGM blood (65%) and breast milk (75.8%) memory B-cell populations. There was a similar IgG isotype frequency of Env-reactive (84.1%) and non-reactive antibodies (66.7%) in this species (Fig. 4a). This finding is consistent with previous reports of HIV Env-reactive monoclonal antibodies isolated from breast milk B cells of HIV-infected women being predominantly IgG isotype[24]. Of the total monoclonal antibodies isolated from AGMs, $V_H 4$ (47.5%) and $V_H 3$ (32.4%) were the primary $V_H$ families used by memory B-cell antibodies found in blood ($V_H 4 = 47.5\%$ and $V_H 3 = 32.5\%$) and breast milk ($V_H 4 = 47.5\%$ and $V_H 3 = 32.3\%$) in AGMs. There was also a similar proportion of $V_H 4$ and $V_H 3$ between the Env reactive ($V_H 4 = 59.1\%$ and $V_H 3 = 25\%$) and non-reactive antibodies ($V_H 4 = 43.7\%$ and $V_H 3 = 34.8\%$) (Fig. 4b). These antibodies were derived from a diverse range of all $D_H 1 - D_H 7$ and $J_H 1 - J_H 6$ gene segments (Supplementary Fig. 10). We next evaluated the SHM of $V_H$ genes of AGM Env-reactive and non-reactive monoclonal antibodies. No difference was observed between the $V_H$-SHM of Env reactive ($9.5 \pm 3.5\%$, $n = 44$) and non-reactive ($10.0 \pm 5.4\%$, $n = 131$) monoclonal antibodies ($P = 0.8$, exact Wilcoxon test) or within the $V_H$-SHMs among different specificities of SIV Env-reactive antibodies (gp41, gp120 and gp140) ($P = 0.7$, gp120 monoclonal antibodies versus Env cross-reactive monoclonal antibodies; antibodies that

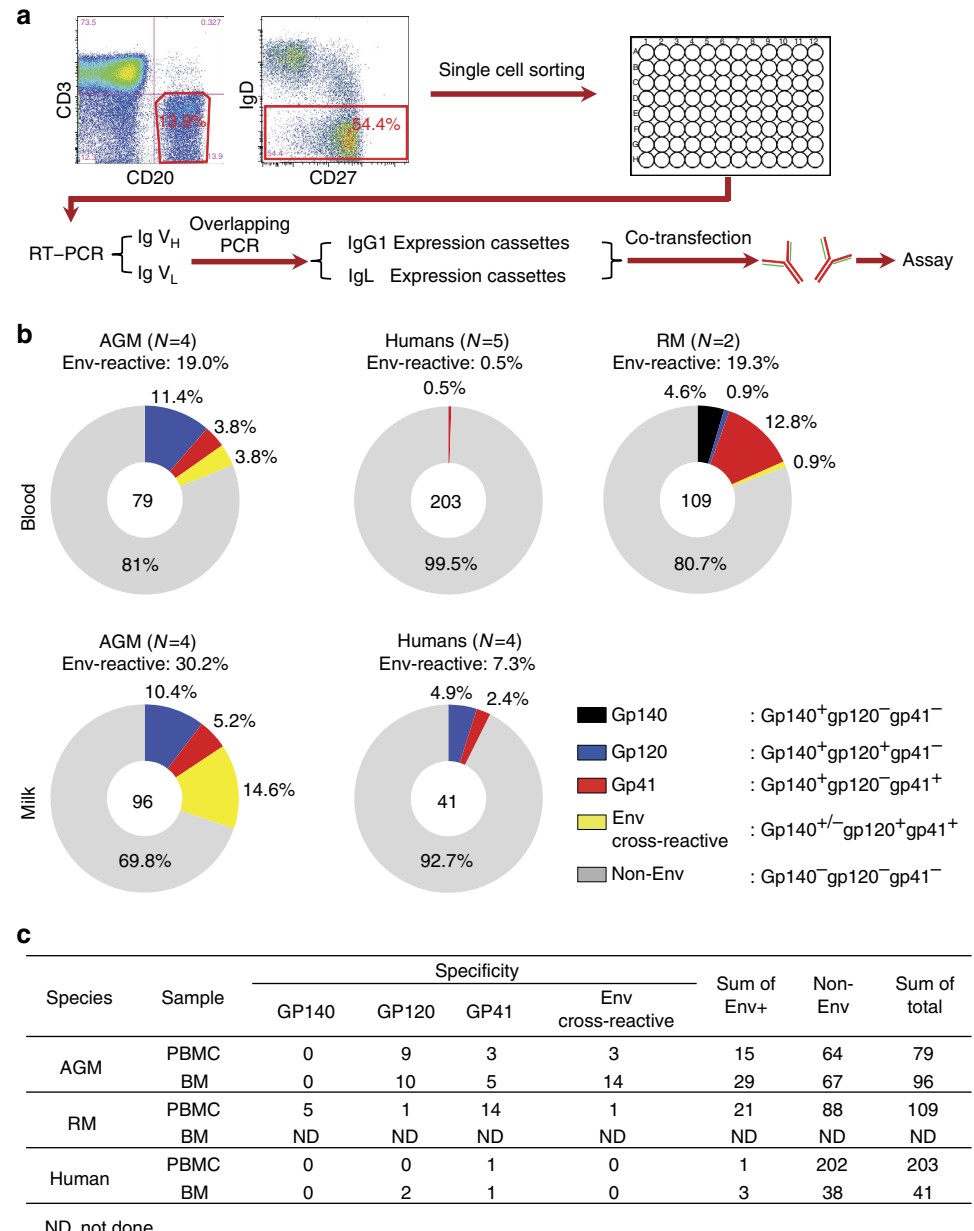

**Figure 3 | High proportion of Env-reactive memory B cells in chronically SIV-infected AGMs compared with that in RMs and humans.** Env reactivity of monoclonal antibodies (mAbs) isolated from single-cell sorted memory B cells of chronically SIV/HIV-infected AGM, RM and human blood and breast milk. (**a**) Schematic presentation of mAb isolation process. Memory B cells were sorted into 96-well PCR plates as CD3⁻CD20⁺IgD⁻CD27^all for AGM and RM samples and CD3⁻CD19⁺IgD⁻CD27^all for human samples. V(D)J gene segments were then isolated from single memory B cell by RT reaction followed by nested PCR and assembled into linear Ig gene expression cassettes of heavy and light chain for the production of recombinant mAbs. (**b**) Total numbers of mAbs isolated from blood of AGMs ($N = 4$), RMs ($N = 2$) and humans ($N = 5$) are indicated at the centre of each pie chart. Percentages of mAb binding to gp140, gp120, gp41, or multiple antigens are indicated in different colours. AGM blood mAbs (19.0%) were SIV Env-reactive, which was significantly higher than those isolated from human blood (0.5%, $P < 0.0001$, Fisher's exact), but not different from those isolated from RM blood (19.3%, $P = 1.0$, Fisher's exact). Yet, the percentage of gp120-specific mAbs in AGM blood (11.4%) was significantly higher than that in RM (0.9%, $P = 0.001$, Fisher's exact). Of 96 mAbs isolated in AGM breast milk ($N = 4$), 30.2% were SIV Env-reactive, which was significantly higher than that in human breast milk ($N = 4$, 7.3%, $P = 0.01$, Fisher's exact). The proportion of gp120-specific mAbs isolated from milk memory B cells was not significantly different in AGM (10.4%) and human breast milk (4.9%, $P = 0.7$, Fisher's exact). Gp140 indicates mAbs that bound gp140 only; Gp120 indicates mAbs that bound gp120, bound or not bound gp140; gp41 indicates mAbs that bound gp41, bound or not bound gp140; Env cross-reactive indicates mAbs that bound gp140, gp120 and gp41; or gp120 and gp41; Non-Env indicates mAbs that did not bind Env protein tested. (**c**) Summary of Env-binding distribution of mAbs isolated from SIV-infected AGMs, RMs and HIV-infected human subjects.

bind to both gp120 and gp41, $P = 0.07$, gp120 monoclonal antibodies versus gp41 monoclonal antibodies; $P = 0.5$, gp41 monoclonal antibodies versus Env cross-reactive monoclonal antibodies; all by exact Wilcoxon test) (Fig. 4c). Similarly, we found no difference in the HCDR3 lengths between Env-reactive and Env non-reactive monoclonal antibodies ($P = 1.0$, Kolmogorov–Smirnov test) (Fig. 4d) or among different specificities of Env-reactive antibodies (gp41, gp120 and gp140

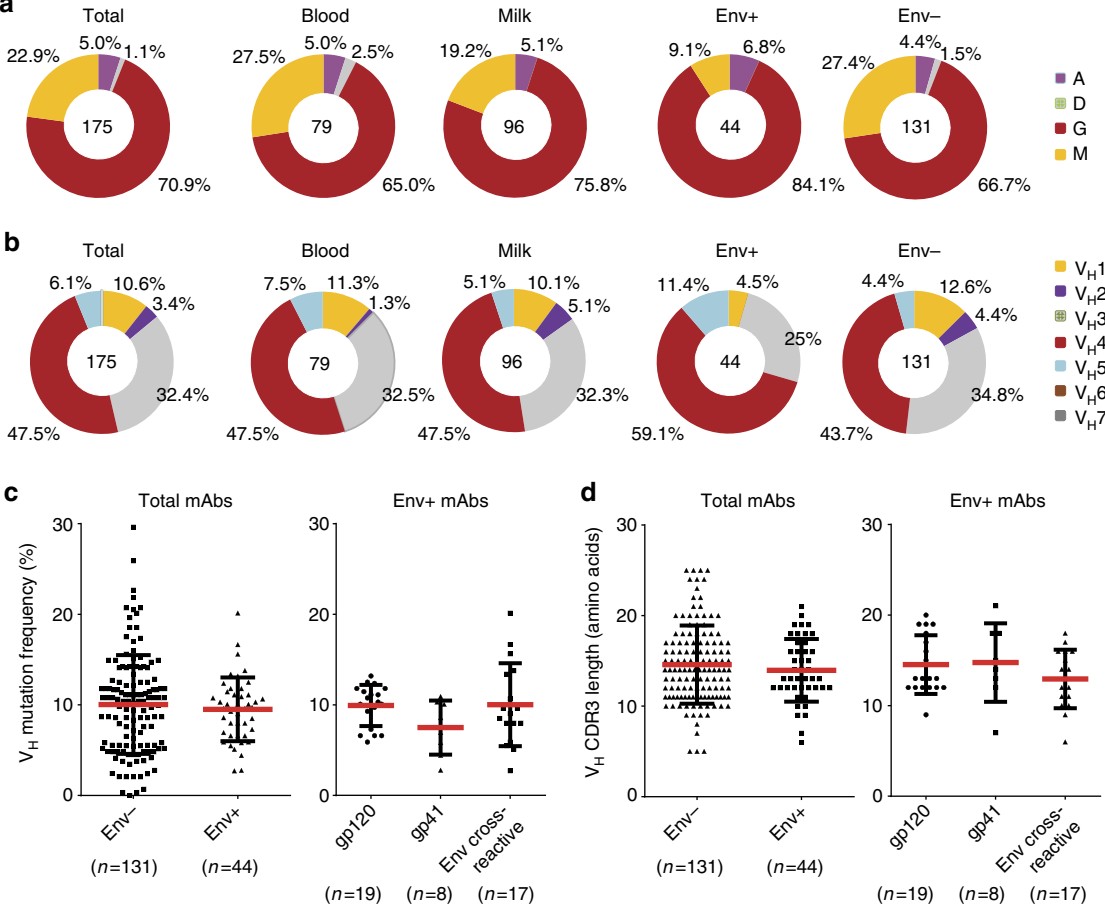

**Figure 4 | Genetic characterization of monoclonal antibodies (mAbs) isolated from single-cell sorted memory B cells of chronically SIV-infected AGMs.** Isotypes and $V_H$ families determined by alignment with human germ lines are shown in pie charts. Numbers of antibodies were indicated at the centre of each pie chart. Percentage of each isotype or individual $V_H$ gene families of the isolated antibodies isolated from milk and blood including Env-reactive and non-Env-reactive antibodies are indicated. (**a**) IgG isotype was dominant both in blood (65%) and milk (75.8%) memory B cells, and over-represented in Env-specific memory B cells (84.1%). (**b**) $V_H4$ and $V_H3$ were dominant $V_H$ families and used AGM in both Env-specific and non-Env-specific mAbs. (**c**) $V_H$ mutation frequency of Env-specific, non-Env-specific and Env epitope-specific AGM mAbs. (**d**) HCDR3 length of Env-specific, non-Env-specific and Env epitope-specific AGM mAbs. There was no difference in $V_H$ mutation frequency and HCDR3 length among Env-specific and non-Env-specific, or Env epitope-specific AGM mAbs. SHM frequency comparisons were made by an exact Wilcoxon rank sum test and HCDR3 lengths were made by a Kolmogorov–Smirnov test.

(antibodies that bound to gp140, but not gp120 or gp41) ($P = 0.6$, gp120 versus Env cross-reactive, $P = 1.0$, gp120 versus gp41, $P = 0.8$, gp41 versus Env cross-reactive) (Fig. 4d).

**SIV Env-specific IgG mAb epitope specificity and function**. To further characterize the epitope specificity and function of SIV Env-reactive antibodies in AGMs, we isolated ten AGM $V_H$ and $V_L$ gene pairs that demonstrated (Supplementary Fig. 11) the highest magnitude binding profile to gp120 ($n = 7$) and gp41 ($n = 3$) in screening enzyme-linked immunosorbent assays (ELISAs) and produced them as recombinant monoclonal antibodies for further characterization (Table 1 and Supplementary Data 4) (two gp120-specific and three gp41-specific monoclonal antibodies from breast milk, and five gp120-specific monoclonal antibodies from blood). We assessed the fine epitope specificity of these Env-reactive monoclonal antibodies via binding to a panel of autologous SIVsab92018ivTF recombinant Env proteins and peptides including SIVsab92018ivTF gp120, gp140, variable loop 1 and 2 (V1V2), variable loop 3 (V3) and sp400 linear peptides[10]. Of the five Env-specific monoclonal antibodies isolated from blood, one was V3 specific, whereas the others were gp120

specific without further epitope specificity identified. Of the monoclonal antibodies isolated from breast milk of SIV-infected AGMs, two monoclonal antibodies were gp120 specific, one was V3 specific and the remaining three were gp41/gp140 specific, which was defined by binding to SIVsab Env gp140 but not to gp120. Furthermore, all of the gp41/gp140-specific milk antibodies bound strongly to the linear immunodominant sp400 region of gp41 (Table 1).

We next examined the antiviral functions of those ten representative high-binding gp41 and gp120-specific Env-reactive monoclonal antibodies isolated from the SIV-infected AGM memory B-cell pool (Table 1). To investigate their neutralizing activity, we tested the AGM monoclonal antibodies against the autologous challenge virus SIVsab92018ivTF and a neutralization-sensitive, tier 1A SIVmacTCLA Env pseudo virus in a TZM-bl reporter cell neutralization assay. We found that none of these ten monoclonal antibodies neutralized the autologous SIV strain ($IC_{50} > 50 \mu g\,ml^{-1}$) (Table 1). However, two blood gp120-specific monoclonal antibodies, DH549 and DH551, neutralized the tier 1A virus with strong potency ($IC_{50} < 0.02 \mu g\,ml^{-1}$) (Table 1). To then examine the non-neutralizing antiviral functions of AGM Env-specific

**Table 1 | SIV Env reactive mAbs isolated from chronically SIV-infected AGMs.**

| Ab ID | $V_H$ | HCDR3 (AA) | H Mut (%) | Isotype | $V_L$ | LCDR3 (AA) | L Mut (%) | Sample Type | EC$_{50}$ ($\mu$g ml$^{-1}$) | | | | | Neutralization (IC$_{50}$, $\mu$g ml$^{-1}$) | |
|---|---|---|---|---|---|---|---|---|---|---|---|---|---|---|---|
| | | | | | | | | | gp120 | V1V2 | V3 | gp140 | sp400 | SIVmacTCLA | SIVsab92018 |
| DH543 | 5–13 | 12 | 6.6 | G | $V_\lambda 1$–73 | 11 | 7.1 | Milk | >100 | >100 | >100 | **0.009** | **0.053** | >50 | >50 |
| DH544 | 3–22 | 11 | 8.5 | G | $V_\lambda 1$–57 | 11 | 5.2 | Milk | >100 | >100 | >100 | **0.005** | **0.087** | >50 | >50 |
| DH545 | 4–152 | 9 | 15.6 | G | $V_\kappa 2$–92 | 9 | 7.5 | Milk | >100 | >100 | >100 | **0.003** | **0.07** | >50 | >50 |
| DH546 | 4–152 | 19 | 11.8 | G | $V_\lambda 1$–73 | 11 | 9.4 | Milk | **0.37** | >100 | >100 | **0.021** | >100 | >50 | >50 |
| DH547 | 1–93 | 19 | 11.8 | G | $V_\lambda 1$–73 | 12 | 4.1 | Milk | **0.054** | >100 | **0.497** | **0.005** | >100 | >50 | >50 |
| DH548 | 4–76 | 15 | 12.4 | G | $V_\lambda 3$–36 | 11 | 5.0 | Blood | **0.004** | >100 | >100 | **<0.001** | >100 | >50 | >50 |
| DH549 | 3–67 | 13 | 5.9 | G | $V_\kappa 1$–NL2 | 9 | 6.1 | Blood | **0.001** | >100 | >100 | **0.002** | >100 | **<0.02** | >50 |
| DH550 | 3–67 | 12 | 12.5 | G | $V_\kappa 3$–53 | 9 | 4.2 | Blood | **0.002** | >100 | >100 | **0.002** | >100 | >50 | >50 |
| DH551 | 4–88 | 19 | 13.2 | G | $V_\kappa 1$–39 | 9 | 3.8 | Blood | **0.004** | >100 | 2.229 | **0.003** | >100 | **<0.02** | >50 |
| DH552 | 4–88 | 17 | 7.3 | G | $V_\lambda 1$–73 | 11 | 2.6 | Blood | 5.899 | >100 | >100 | **0.397** | >100 | >50 | >50 |

AGM, African green monkey; HCDR, heavy chain complementarity-determining region; LCDR, light chain complementarity-determining region; mAb, monoclonal antibody; SIV, simian immunodeficiency virus; TCLA, T-cell line-adapted; $V_H$, heavy chain variable; $V_L$, light chain variable.
High binding strength (EC$_{50}$) and potent neutralizing activity (IC$_{50}$) are denoted in bold.

monoclonal antibodies, we measured their ability to capture the whole autologous SIV challenge virus SIVsab92018ivTF virion in the presence or absence of soluble CD4 molecule (sCD4). We found that in the presence of sCD4, five of the seven gp120-reactive antibodies captured high levels of virions, suggesting that these antibodies may target the gp120 epitopes that are better exposed on CD4 engagement (Fig. 5a). However, none of the three gp41-reactive antibodies strongly captured virions in the presence or absence of sCD4 as compared with the positive control HIV anti-gp41 monoclonal antibody 7b2.

To assess the contribution of antibody-dependent cellular cytotoxicity (ADCC)-mediating monoclonal antibodies to plasma ADCC responses in SIV-infected AGMs, those monoclonal antibodies were then tested for ADCC activity against SIVsab92018ivTF gp120/140-coated CD4$^+$ T cells and for the ability to bind cells infected with SIVsab92018ivTF. All ten Env-specific monoclonal antibodies were capable of mediating ADCC (Fig. 5b,c). Out of the seven gp120-specific antibodies, five (DH548, DH546, DH550, DH549 and DH552) mediated robust ADCC against gp120 and g140 Env-coated cells with endpoint concentrations of ~0.01 $\mu$g ml$^{-1}$ (Fig. 5b,c). Interestingly, DH546 and DH552 blocked binding to SIVsab92018 gp120 against each other with at least 20% blocking activity as measured by a blocking ELISA previously described[29] (Supplementary Table 5). DH550 and DH549 also blocked binding against each other, suggesting that these gp120-specific ADCC-mediating antibodies target overlapping epitopes on the autologous virus. All gp41 and gp120-reactive monoclonal antibodies were able to bind SIVsab92018ivTF-infected CD4$^+$ target cells, although higher levels of binding were observed for gp120-specific monoclonal antibodies. In particular, those monoclonal antibodies that demonstrated the most potent ADCC (DH548, DH546, DH550, DH559 and DH552) were also best at recognizing the SIV-infected cells (Fig. 5d). Together, these results suggest that AGM SIV Env-reactive antibodies can mediate robust non-neutralizing antiviral functions.

## Discussion

In this study, we identified and characterized Ig loci from the recently published AGM genome[13]. We found that AGM Ig heavy and light-chain germline genes are similarly structured and highly homologous with those of RMs and humans, albeit more homologous to RMs than to humans (Fig. 1 and Supplementary Figs 4 and 5). Yet, we identified only two IgG and one IgA constant regions from this assembled version of the AGM

genome, compared with four IgG and two IgA constant regions in humans[17]. Considering the predicted large segment missing in $V_H$ locus (Supplementary Fig. 6), and the fact that only constant domain 2 ($C_H2$) and $C_H3$ were identified in the AGM *IgG2* gene, it is possible that other large segments were also missing in heavy constant gene loci in the current version of AGM genome. In addition, it should be cautioned that as this newly established $V_H$ and $V_L$ gene database is from a single AGM, we might have missed $V_H$ germline genes and potential intra-species allelic variations. Thus, definition of the AGM Ig loci would benefit from improved AGM draft genome that includes more individual genomes. Nevertheless, despite this potential limitation, we have developed the basic database of AGM Ig variable genes, which now serves as a new platform to interrogate memory B cell populations in AGMs.

We then utilized this platform to genetically characterize 44 SIV-reactive monoclonal antibodies isolated from the blood and milk memory B cells from four chronically SIV-infected AGMs. We found that the predominant isotype of monoclonal antibodies isolated from AGM blood and breast milk memory B cells was IgG. These antibodies were diversely distributed in various $V_H$ gene families, with $V_H4$ and $V_H3$ being the two most predominant $V_H$ families used in both SIV Env-reactive and Env non-reactive antibodies. Antibodies in both SIV Env-reactive and non-reactive antibody groups also had similar distribution of HCDR3 length and SHM frequency (Fig. 4). These results indicate that there was no apparent bias in amplification of AGM Ig variable genes using both RT–PCR primers and conditions initially developed for isolating Ig variable gene RMs[21].

Remarkably, chronically SIV-infected AGMs had a significantly higher proportion of their memory B cells with specificity to SIV Env (19% in blood and 30% in breast milk) compared with the small proportion of HIV Env-specific memory B cells identified in chronically HIV-1-infected individuals (0.5% in blood and 7.3% in breast milk). Furthermore, the predominantly SIV Env-reactive circulating memory B cells in AGMs were directed against gp120 (60% of Env-reactive antibodies), which is in contrast to the high proportion of gp41-directed memory B cells seen in RMs (66.3% of Env-reactive antibodies) (Fig. 3) and in acutely HIV-infected and HIV-vaccinated humans[5,30]. One caveat to this monoclonal antibody analysis is that AGM and RM antibodies were screened with the immunodominant sp400 gp41 linear peptides[9,10] due to a lack of conformational SIV gp41 reagent. Yet, the HIV sp400 was also used to screen human antibodies. Moreover, although there is a clear difference in the proportion of Env-directed memory B cells between AGMs and

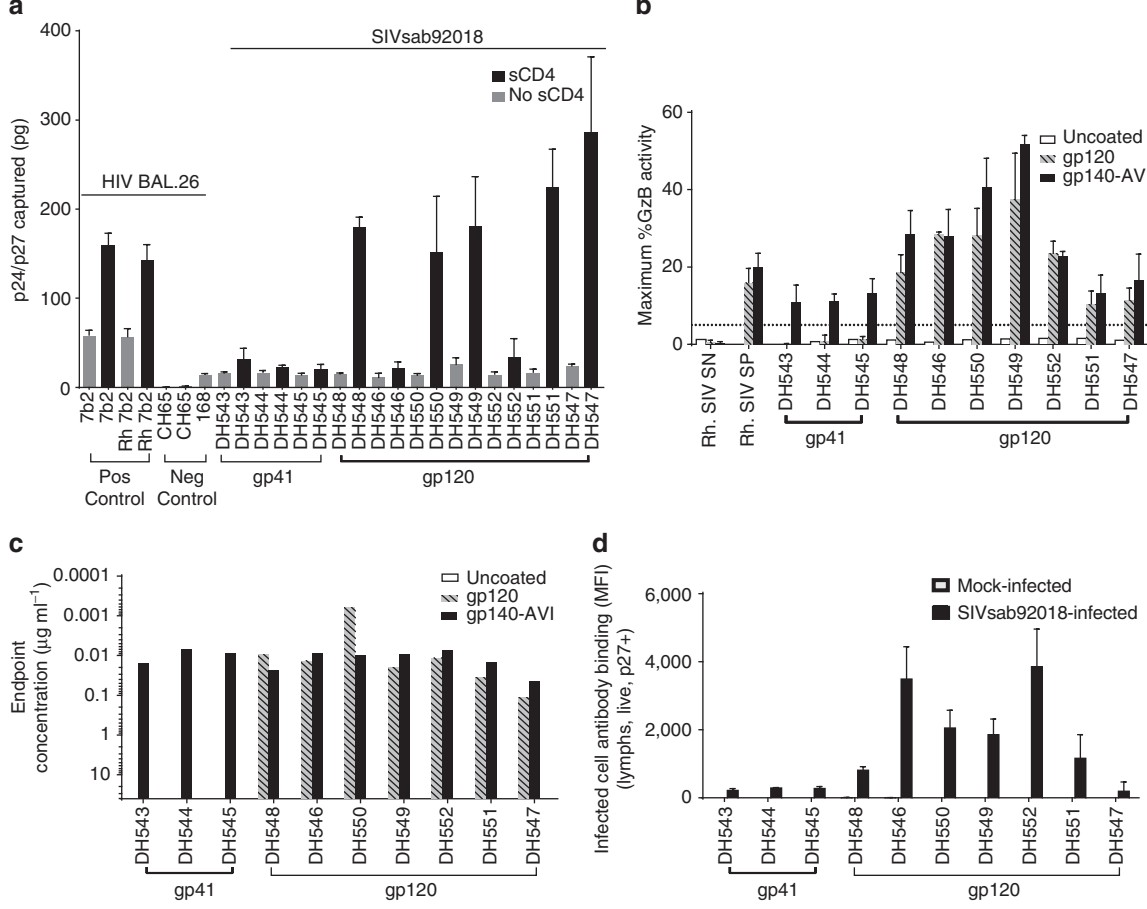

**Figure 5 | Virus capture and ADCC activity of Env-specific AGM monoclonal antibodies (mAbs).** (**a**) MAbs were tested for the ability to capture the autologous challenge virus SIVsab92018ivTF in the presence (black bars) and absence (grey bars) of sCD4. Five of ten SIV gp120-specific mAbs isolated from chronically SIV-infected AGMs robustly captured the autologous challenge virus in the presence of sCD4. MAb7b2 was used as a positive control for capture of HIV-1 Bal.26 and an influenza HA-specific antibody CH65 was used as the negative control. (**b**) SIV Env-specific AGM mAbs were tested for ADCC activity against the autologous challenge virus SIVsab92018ivTF Env gp140- and gp120-coated cells. Data represent the maximum ADCC activity (% Granzyme B (GzB) activity) and the dashed line represents the positivity threshold (5% GzB activity) established using uncoated cells as targets. All gp120-specific mAbs mediated ADCC against SIV Env gp120-coated target cells, whereas all gp41 directed mAbs mediated ADCC against gp140-coated target cells. (**c**) ADCC activity was observed for both SIV gp120- and gp41-specific mAbs with endpoint concentrations of ~0.1–0.01 μg ml$^{-1}$. (**d**) SIV-specific AGM mAbs bound to virus-infected cells. The data in **a,b,d** show means ± s.d.

humans, it should be noted that consensus-S may not detect all HIV Env IgG specificities and potentially underestimate the proportion of Env-specific B cells in humans. Despite this limitation, several studies have demonstrated the ability of consensus-S to detect cross-clade reactive antibodies[30–32].

After co-evolving with the virus for more than 30,000 years, AGMs have developed adaptive mechanisms to limit SIV pathogenesis[33]. Many studies have investigated immune mechanisms contributing to the lack of an AIDS-like disease in natural hosts of SIV[8,34–37]. Although we observed that AGMs had high proportions of SIV Env-reactive memory B cells, previous studies have challenged the contribution of humoral immunity to the lack of SIV disease progression in natural hosts, showing that the depletion of B cells and its resulting decreased antibody responses during acute and chronic SIV infection of AGMs did not demonstrate accelerated disease pathogenesis[37]. Despite the high proportion of Env-specific B cells in chronically SIV-infected AGMs and the demonstration of rapid induction of autologous neutralizing responses[9], it was surprising that the isolated high-binding Env-specific monoclonal antibodies exhibited only limited neutralization activity against a tier 1A SIV variant. Yet, AGM gp120-specific monoclonal antibodies had strong

ADCC and virus capture activity against the autologous SIV challenge strain (Fig. 5). Interestingly, of the seven gp120-specific monoclonal antibodies, two pairs of gp120-specific ADCC-mediating antibodies blocked binding to SIVsab92018 gp120 against each other, suggesting that these gp120-specific AGM monoclonal antibodies target epitopes within the same vicinity in the autologous SIVsab92018 Env. Furthermore, the observation that only five out of ten Env-specific AGM antibodies mediated robust virus capture activity in the presence of sCD4 could be explained by reduced follicular dendritic cell SIV-immune complex trapping and reduced non-conformation epitope-specific B-cell selection in SIV-infected AGM germinal centres[38,39]. It remains unclear whether and how the predominant gp120-specific humoral immune responses in AGMs contribute to the observed disease outcome in this species. Yet, understanding the role of these non-neutralizing antibody-mediated effector functions, such as ADCC, in the natural hosts on viral containment and the low rate of vertical virus transmission in these natural hosts could inform both HIV prevention and viral clearance strategies in humans.

A primary goal of HIV vaccine development is to elicit protective broadly neutralizing antibody responses[2] and/or

mediate ADCC against virus-infected CD4$^+$ T cells[40]. Recently, a large number of bnAbs and ADCC-mediating antibodies have been isolated from chronically HIV-1-infected individuals[41]. Most of the defined epitopes of bnAbs and ADCC antibodies map to the Env gp120 subunit[4,41]. Yet, gp41 is the predominant target of the humoral immune response that develop in HIV-1/SIV infected humans[5] and RMs (Fig. 3), and in Env-immunized humans[6]. Our study reveals that SIV-infected AGMs have a B-cell response that yields a gp120-dominant response during chronic SIV infection (Fig. 3)[9] and this feature is distinct from that of non-natural HIV/SIV hosts. Thus, interrogating the Env-specific memory B-cell response in AGMs may uncover mechanisms of this gp120-focused B-cell response, which may provide new clues for developing novel HIV vaccines that will direct the human B-cell response towards more desirable HIV Env epitope targets.

NHP models are commonly used in HIV immunization studies to address questions relevant to B-cell lineage design, with RMs being the most widely used NHP model. The draft Ig database of RM had been assembled from the complete genome of one rhesus macaque[11]. Moreover, this database has been recently refined with Ig-specific next generation sequencing from ten Indian-origin RMs[12], making it possible to accurately assess HIV vaccine-induced B-cell lineages. Yet, owing to their specific immune features, AGMs may represent a potential new NHP model to evaluate novel HIV vaccine concepts. A key obstacle to use the AGMs as a viable NHP model in HIV and other pathogen vaccine studies was the lack of an Ig germline gene database and an efficient platform to analyse B-cell responses to relevant HIV/SIV immunogens. Our study opens the door to use the AGM NHP model in HIV vaccine studies. This NHP model offers a distinct immune landscape, which co-evolved with a virus similar to HIV-1, and also provides a platform for investigating B-cell responses to other vaccines.

## Methods

**Nonhuman primates and sample collection.** Six female AGMs (*Chlorocebus sabaeus*) and four female RMs (*Macaca mulatta*) between 4 and 11 years of age were hormonally treated to induce lactation as described previously[42]. AGMs and RMs were intravenously inoculated with $7.9 \times 10^8$ copies of cloned SIVsab92018ivTF and $2.1 \times 10^5$ copies of SIVmac251.30 virus stock, respectively[42]. Blood and milk samples were collected at 1 year post infection. Plasma and peripheral blood mononuclear cells and lymphocytes in breast milk samples were isolated as described previously[43]. All animals were housed and maintained according to ref. 44.

**Human subjects.** Peripheral blood and breast milk samples were collected from seven chronically HIV-1-infected individuals at least 200 days after transmission, as estimated from Fiebig classification[45] and patient medical history (Supplementary Table 1)[30]. All human subject work was performed with informed consent from trial participants. All work related to human subjects was in compliance with Institutional Review Board protocols approved by the Duke University Health System Institutional Review Board and the College of Medicine Research and Ethics Committee in Malawi.

**AGM genomic sequence.** The genomic sequence of AGM (*Chlorocebus aethiops sabaeus*, the Caribbean colony and animal ID 1994-021) was recently assembled and released[13]. The online link (https://genomequebec.mcgill.ca/compgen/browser-VPR/cgi-bin/hgGateway) to Genome Browser Gateway and BLAT service were kindly supplied by McGill University and Génome Québec Innovation Center.

**Organization of AGM Ig heavy and light chain gene germlines.** AGM Ig heavy, κ and λ chain loci were located on chromosomes (CHR) 24, 19 and 14, respectively, by aligning human Ig gene constant regions to the AGM genome of *C. sabaeus* 1.0. The sequences of CHR24, 19 and 14 were split into 700 bp DNA fragments with 350 bp overlap and were used to align with human Ig germline genes to identify AGM Ig variable regions (V-region) by using the online alignment tool IMGT/HighV-QUEST[46]. D-regions and J-regions were manually identified by aligning with human D-regions and J-regions in AGM genomic DNA fragments between V-regions and C-regions. Unplaced contigs and unassembled reads of

AGM genome sequences were analysed in the same aforementioned manner. All V, D and J-regions were then manually analysed and classified into functional (F), potential functional ([F]), open read frame (ORF) and pseudogene (P) adopted from IMGT[47]. Briefly, the functional germline genes have an open reading frame without a stop codon, no described defect in splicing sites, rearrangement signal sequences, translation initiation codon (ATG) in the leader part I of V-region and conserved amino-acid residues. The germline genes that met these criteria and those that did not have a classic heptamer sequence of RRS were defined as potential functional genes [F]. Germline genes without in-frame stop codon or frame shifting, without canonical ACCEPTOR-SPLICE, were identified as an ORF gene. Orphan genes out of the key chromosome were also classified as ORF genes. All remaining germline genes were identified as the pseudogenes.

The relative chromosomal position of the Ig germlines was determined by alignment to AGM genome of *C. sabaeus* 1.0 using a BLAT server (http://www.ncbi.nlm.nih.gov/assembly/GCA_000409795.1). All Ig germline genes were named according to IMGT nomenclature[19]. Names of all AGM V-regions were adapted from the subgroup nomenclature of the closest similar human V-region. The D-regions and J-regions were sequentially named according their chromosomal position. The phylogenetic relationship of AGM $V_H$ genes was then evaluated by maximum likelihood construction using MEGA5.2 software and compared with the $V_H$ genes of RMs[11] and humans[14]. The complete physical map and annotation of the Ig germline genes were drawn (Supplementary Figs 1, 7 and 8).

**Single-cell sorting by flow cytometry.** Total memory B cells of three species were sorted by flow cytometry as described previously[21]. Total memory B cells were gated as CD3$^-$CD20$^+$IgD$^-$CD27$^{all}$ for AGM ($n=4$) and RM ($n=2$) samples, and CD3$^-$CD19$^+$IgD$^-$CD27$^{all}$ for human samples ($n=7$).

**Transient production and specific screening of immunoglobulin heavy and light chain genes.** Ig V(D)J rearrangements from human samples were isolated by nested RT–PCR using the protocol as previously described[22]. To avoid primer mismatch V gene sequences including the framework region with high SHM as typically observed for HIV-1 broadly neutralizing antibodies[48], we designed both the external and internal forward primers on the 5' leader V-region for AGM and RMs (Supplementary Table 3). Isolation of Ig V(D)J from AGM and RM samples was carried out using a modified nested RT–PCR protocol[21]. Isolated $V_H$ and $V_L$ genes were used for assembling full-length Ig IgG1 heavy- and light-chain linear expression cassettes by overlapping PCR to express recombinant IgG1 antibodies (Supplementary Fig. 12) and screened for Env-binding as described previously[22].

**Production of recombinant monoclonal antibodies.** Ten AGM antibody genes with the strongest binding profile to gp120 ($n=7$) and gp41 ($n=3$) were *de novo* synthesized and cloned (Genscript, Piscataway, NJ) into plasmids containing rhesus IgG/IgK/IgL constant regions. Recombinant monoclonal antibodies were produced and purified from 293F cells (Life Technologies, Grand Island, NY) by co-transfecting heavy- and light-chain plasmids as described previously[21].

**Sequence analysis and SHM estimation.** Sequences were initially analysed as described previously[49]. Briefly, sequences were aligned with consensus human $V_H$ and $V_L$ genes using a trace-back path for each $V_H$ and the position of the invariant cysteines is determined. Functional sequences were annotated with AGM germline genes to calculate the SHM and infer VDJ rearrangement. SHM was calculated using the formula as the following:

SHM (%) = (mutated nucleotides in V-region/V-region nucleotides) × 100%

V-region in calculation inferred the regions from start of Framework 1 through the end of Framework 3.

**SIV and HIV-1 Env ELISA.** SIVsab92018ivTF gp120, gp140, scaffolded V1V2 protein, V3 peptide (5'-KTVLPVTIMAGLVFHSQKYNT-3') and gp41 immuno-dominant region peptide (sp400 seq: 5'-RVTALEKYLEDQARLNIWGCAFRQI CHTTVPWKFNNTPDWNN-3') were used to screen AGM and RM monoclonal antibodies in an ELISA. HIV-1 Cons gp140, Cons gp120, MN gp41 and sp400 were used for antibodies isolated from humans by ELISA as described previously[50]. Positivity binding cutoffs were set as three times the binding of the mock transfection reactions in AGM, RM and humans.

**Neutralization assay.** AGM monoclonal antibodies against SIV variants was measured in TZM-bl cell line as described previously[51]. Neutralization activity of monoclonal antibodies was reported as 50% reduction in relative luminescent units as compared with the virus control after the subtraction of the background relative luminescent units from the cell control. The autologous, SIVsab92018ivTF infectious molecular clone and tier 1A T-cell line-adapted Env pseudovirus (SIV mac239 backbone) were tested against all monoclonal antibodies[52,53].

**Virus capture assay.** Purified AGM monoclonal antibodies (7 gp120 and 3 gp41 monoclonal antibodies) were tested for virion capture in the presence and absence

of sCD4 (gift from Bing Chen) at 37 °C for 1 h. Virus was lysed overnight at 4 °C with 0.1% Triton X-100. The p24 and p27 quantification was performed with a Perkin-Elmer and Zeptometrix kit, respectively. Virus capture activity was calculated as the amount of p24/p27 in $pg\,ml^{-1}$ of the input virus volume by standard curve interpolation.

**ADCC assay.** ADCC activities of purified AGM monoclonal antibodies were detected by GranToxiLux procedure against $CEM.NKR_{CCR5}$ target cells[54] as described previously[55]. ADCC endpoint concentrations were determined by interpolating concentrations of monoclonal antibodies that intersect the positive cut-off of 5% Granzyme B activity using Graph Pad Prism 5 software (Graph Pad, La Jolla CA).

**Monoclonal antibody binding to Env on the surface of SIV-infected CD4$^+$ cells.** Indirect surface staining[56] was used to test binding of purified AGM monoclonal antibodies to $CEM.NKR_{CCR5}$ CD4$^+$ T cells infected with SIVsab92018iv TF infectious molecular clone virus using DEAE-Dextran as described previously[55]. The monoclonal antibodies were tested at $10\,\mu g\,ml^{-1}$ and binding was measured after a 2-h incubation with the infected target cells at 37 °C.

**Cross-blocking ELISA of ADCC-mediating antibodies.** 384-Well ELISA plates were coated with SIVsab92018gp120 and blocked with assay diluent (PBS containing 4% (wt/vol), whey protein, 15% normal goat serum/0.5% Tween 20) for 1 h at room temperature. Purified AGM gp120-specific monoclonal antibodies that mediated robust ADCC activity against the challenge virus were biotinylated and tested for their ability to block one another at $100\,\mu g\,ml^{-1}$ against SIVsab92018gp120. Biotinylated monoclonal antibodies were detected with streptavidin horseradish-peroxidase at 1:30,000, washed and followed by substrate. Reactions were stopped by the addition of 1% HCl and read at 450 nm. Per cent blocking was calculated as follows: 100 − (purified AGM monoclonal antibody mean optical density/negative control mean optical density) × 100.

**Statistical analysis.** Per cent homology comparisons between AGMs and humans versus AGMs and RMs were made using the Wilcoxon test. Monoclonal antibody Env-positive and gp120 Env-specific comparisons between AGMs and other species by compartment were made using Fisher's exact tests. Monoclonal antibody Env-positive and gp120 Env-specific comparisons within AGMs were made using the Kolmogorov–Smirnov test for CDR3 length, and Wilcoxon and exact Wilcoxon for VH mutation rate. All raw $P$-values for the tests were assembled and corrected for family wise alpha inflation using the Benjamini–Hochberg correction[57]. All statistical tests were completed using SAS v9.4 (SAS Institute Inc., Cary, NC, USA) (Supplementary Table 4).

**Data availability.** The Ig gene sequences have been deposited in the GenBank Nucleotide database with accession codes KX264380 to KX264399. The authors declare that all other data supporting the findings of this study are available within the article and its Supplementary Information files or from the corresponding author upon request.

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

## Acknowledgements

This work was supported by NIH awards K08AI087992-04 and R21AI100760-01 (SP), and the grant from the NIH, NIAID and AI067854 (the Center for HIV/AIDS Vaccine Immunology). We thank Dawn Marshall for assisting with the sorting of single memory B cells of RM and AGM samples; and Tom Kepler for his assistance with data interpretation.

## Author contributions

R.J. identified and annotated the AGM Ig genes, performed B cells analysis and drafted the manuscript. D.M. and Q.N. performed the monoclonal antibody assays and analysis. J.P. and R.W.E. performed the ADCC and infected cell binding analysis. T.A. performed the monoclonal antibody isolation. C.S. and A.F. performed the monoclonal antibody screening. M.W. performed the variable gene analysis. A.T. contributed to the human monoclonal antibody analysis and data interpretation. J.A. performed flow cytometry sorting and analysis. J.E.H. produced the monoclonal antibodies. N.V. performed the statistical analysis. L.C. performed the non-human primate procedures. K.D., N.J. and J.W. performed the AGM genome sequencing and annotation. B.H. contributed to data analysis and interpretation. G.F. directed the ADCC analysis and contributed to data interpretation. H.L. and S.P. conceived and designed the study, oversaw the data analysis and interpretation, and wrote the manuscript.

## Additional information

**Competing financial interests:** The authors declare no competing financial interests.

