## [Peer review file · Nature Communications]

Reviewers' comments:

Reviewer #1 (Remarks to the Author):

The manuscript by Zhang et al titled "the immunoglobulin repertoire of chronically simian immunodeficiency virus infected African green monkeys" is really quite an impressive body of work. The authors define the features of the AGM immunoglobulin loci and utilize this information to understand and compare the virus-specific memory B cell responses of AGMs to both humans and SIV-infected rhesus macaques. The authors denote some interesting differences in the proportions of SIV/HIV-specific responses in AGMs compared to SIV-infected RMs and humans, with AGMs having a higher proportion of Env-specific responses. The authors went further to try to understand the effector function of these Env-specific Abs in AGM and found that while they were poor at neutralizing virus they were robust at mediating ADCC and CD4-dependent virion capture. This review appreciates the enormous effort and detail that went into this work and only has minor comments and/or concerns listed below.

1. While there appears to be significant differences in the proportional representation of Env-specific Ab responses in AGMs compared to RMs and humans, were these statistically significant? There appears to be an absence of formal statistical measures throughout the manuscript that either needs to be changed or explained.
2. While the proportions of Env-specific Ab responses in AGMs was higher than RMs and humans was there a quantitative difference between the species? Did AGMs simply have a weaker response in which a higher proportion were Env-specific, or robust responses with higher proportions being Env-specific?
3. The ADCC experiments certainly demonstrated increased effector function by this measure in AGMs on a per Ab level, however, was there any evidence in vivo for this robust effector function (i.e. activated NK cells, increased viral evolution or antibody evolution in the host, etc)?
4. It is interesting the CD4-dependent virion capture was robust in AGMs. Does this mean that non-CD4 mediated virion capture was not? Do the authors believe this may reflect the reduced follicular dendritic cell SIV-immune complex trapping reported in AGMs? This might merit some discussion.

Reviewer #2 (Remarks to the Author):

Summary:

Zhang et al describes the annotation of the immunoglobulin repertoire for African Green Monkeys. AGMs are of interest to HIV research because they are natural hosts for SIV-infection, but do not succumb to disease. The authors show differences in VDJ genes between AGMs, rhesus macaques (RMs), and humans. The authors then isolate monoclonal antibodies, which show stronger immunogenicity to gp120 than to gp41 for AGMs compared to humans or RMs. The HIV Ag-reactive monoclonal antibodies do not show any differences in levels of SHM or CDR3 lengths compared to the non-HIV Ag reactive antibodies. The monoclonal antibodies are also non-neutralizing, but do exhibit ADCC activity.

General comments:

The manuscript is somewhat cumbersome to read and the presentation of data does not fit a

coherent message. The annotation of the AGM IgG is interesting and potentially useful. These data by themselves may not be sufficient to warrant publication in this journal, so it is understandable that subsequent experiments were performed to compare AGM responses to RM and humans. The results of these experiments, however, are vague and the conclusions are overinterpreted.

First, the rationale for AGMs as an alternative model for vaccine studies needs to be better articulated. If SIV-infection of AGMs does not lead to disease progression, why should they be used for vaccine studies? What evidence exists that disease control is antibody mediated (the authors actually list a series of references that would suggest that adaptive immunity is not important for disease control)? Highlighting differences in immune responses between humans and AGMs contradicts the use of these animals as vaccine models for expected responses in humans. There needs to be clearer goals for how or why the AGM antibody repertoire is of use to HIV vaccine design, otherwise the manuscript should be rewritten more simply as a description of the AGM antibody repertoire.

Second, what was the rationale for not antigen sorting memory B cells? Could there be neutralizing antibodies present if the repertoire was sampled more deeply?

Third, is the general conclusion that non-neutralizing/ADCC-active antibodies in AGM might be responsible for disease control? If so, are the antibodies elicited in RM in the same study also non-neutralizing/ADCC-active? Would such a result in RM then negate this hypothesis for AGM?

Finally, it would be interesting/informative to perform competition assays between the monoclonal antibodies isolated from AGM to determine if they're binding the same epitope. Indeed, if AGMs are being championed as an alternative vaccine model, a better comparison of AGM vs RM repertoires and responses needs to be performed. If the argument is that these animal models have different antibody repertoires and that this is the reason for disease control, then immunization of AGMs and RM with the same gp120 protein antigen and the elicitation of different epitope responses would be more convincing to support this argument.

Specific comments:

1. Would be easier to comment on specific items if line numbers were used
2. In the abstract, "Here we define features of the AGM Ig loci and compare their virus-specific memory B cell repertoires to that of humans and RMs" - based on the figures and text, the memory B cells were not antigen sorted?
3. The line, "Although more than half of chronically HIV infected individuals develop some degree of broadly neutralizing antibodies (bnAbs)" is inaccurate, the paper reports broadly neutralizing serum responses, not bnAbs
4. The lines, "Defining the roadmap for how epitope-specific bnAbs are produced will be important in designing potential strategies to induce broadly reactive HIV antibodies. Yet, defining this roadmap for eliciting gp120-directed bnAbs may require unique non-human primate models and B cell repertoire analysis tools." - how does the last sentence lead to the first sentence?
5. The line, "Together, these results suggest that SIV Env-reactive antibodies, particularly those that are gp120-specific, in AGMs may mediate in vivo antiviral function through non-neutralizing mechanisms that may be dependent on Env-CD4 engagement" is more speculation than a conclusion
6. The line, "We then utilized this platform to genetically characterize 175 mAbs isolated from the blood and milk memory B cells from four chronically SIV-infected AGMs." - is somewhat misleading please specify the numbers of mAbs that were Env-reactive (i.e. 44)
7. The line, "Thus, interrogating the Env-specific memory B cell repertoire in AGMs may uncover

mechanisms of this distinctly focused B cell response which may provide new clues for developing novel HIV vaccines that will direct the human B cell repertoire towards more desirable HIV Env epitope targets." - how specifically would evaluating the Env-specific B cell repertoire in AGM lead to this?

Reviewer #3 (Remarks to the Author):

The authors have identified and characterized the functional immunoglobulin repertoire in African Green Monkeys (AGM) using data from the recently published AGM genome. AGM is a natural host species for simian immunodeficiency virus (SIV). This manuscript clearly establishes that AGM Env-specific mBCs are predominantly reactive against gp120 in contrast to the more frequent gp41-directed responses of rhesus macaques and humans. These gp120-reactive mBCs in chronically SIV-infected AGMs exhibit surprisingly limited activity against non-autologous, variant SIV, yet are potent in ADCC and virus-capture activity against the autologous SIV challenge strain; as the authors suggest, a continued study of the role of these non-neutralizing antibody-mediated effector functions in a natural host could shed light on HIV clearance strategies in humans. The conclusions of this novel manuscript are appropriately discussed, and caveats included (e.g., the AGM genome is only a draft and is derived from a single monkey). Statistical treatment of the data (Wilcoxon tests and Fisher's Exact) are appropriate. Zhang et al. are experts in the field and have created a uniquely valuable data set which will allow for detailed immunogenetic investigations of SIV infection in AGM and potentially lead to new insights that may have bearing on gp120-directed HIV vaccine development.

Reviewer #4 (Remarks to the Author):

The paper by Zhang et al. investigates the memory B cell compartment in chronically SIV-infected African green monkeys, which are natural hosts of (non-pathogenic) SIV infection. The approach taken is to isolate monoclonal antibodies (Mabs) from total IgD negative memory B cells and, following cloning and expression of antibody heavy and light chains, the antibody specificities were tested against various forms of Env.

The authors conclude that "AGM Env-specific memory B cells were predominantly gp120-directed in contrast to the frequent gp41-directed responses of RMs and humans" and that co-evolution of SIV and AGMs has resulted in an SIV Env-specific B cell repertoire that differs from that observed in non-natural SIV/HIV infection. These are sweeping statements based on very little amount of data. Upon inspection of the results there are some major concerns that severely cripple the manuscript and make the conclusions drawn invalid. There are also several technical issues, as follows:

1. The title, "The immunoglobulin repertoire of chronically simian immunodeficiency virus infected African green monkeys", is misleading as the authors do not perform deep sequencing of large number of SIV-specific B cells. The study is focused on envelope-specific B cells and very few monoclonal antibodies are isolated and analyzed.

2. In Fig 2, the authors show a phylogenetic tree of rhesus macaque VH genes consisting of 61 different genes. However, the authors refer to a paper by Francica et al, which describes over 100 rhesus macaque V genes. Thus, there seems to be a mismatch in the numbers between what is referenced and what is shown, why is this? Which of these are actual genes and which are allelic variants of the same gene? How many animals were sequenced to define the database published by Francica et al? This should be clarified.

3. The authors compile a database for AGM VH genes based on reference 13. How many animals were analyzed in ref 13? Were the germline versions of all the V genes in the AGM database shown to be expressed in naïve B cells of the AGMs used in the current study? Did the authors look at this?

4. The way the percentages in Figure 3 are calculated is confusing in that percentages of antibodies that bind gp140 are added to the percentages of antibodies that bind gp120 and gp41. These specificities should be largely overlapping and they are therefore "counted twice". This should be revised and the data replotted.

5. The fraction referred to as multi-Env is also confusing. According to the figure legend, this indicates antibodies that bind both gp41 and gp120, which does not make sense. This fraction is also added to the other specificities so again, specificities are counted more than once. Please clarify the intention behind this.

6. In the description of the ELISAs, it says that gp41 reactivity of some Mabs was measured against a linear peptide, rather than against folded protein. This is not ideal as all gp41 reactivities would not be detected this way. This should be commented on.

7. In the ELISAs used to determine the specificities of anti-HIV-1 Mabs, consensus Env proteins were used. While this may be the best one can do given that these individuals harbor diverse virus populations, Cons probes are not going to pick up all reactivities. This may lead to an underestimation of the frequencies of Env-specific memory B cells, which should also be commented on.

8. Importantly, because total memory B cells were sorted rather than using specific probes to pull out Env-specific memory B cells, the number of specific Mabs the authors were able to isolate and analyze is very low. Especially if some of the MAbs are counted more than once. Removing the Multi-Env MABs that also bind other probes the data in Figure 4 demonstrate that only 12 Mabs were isolated from AGM blood and 15 from AGM milk, while from humans only 1 Mab was isolated from blood and 3 from milk. Thus, from five humans included in the study, only a total of 3 Env-specific antibodies were isolated. These are far too low numbers to base any conclusions on. Based on Figure 4, the total number of Mabs appears to be:

From 4 AGM: AGM (Blood): gp120 (9), gp41 (3), multi-Env (3) > total Mabs: 15 (12 if excluding multi-Env); AGM (Milk): gp120 (10), gp41 (5), multi-Env (14) > total Mabs: 29 (15 if excluding multi-Env)
From 5 humans: Human (Blood): gp41 (1); Human (Milk): gp120 (2), gp41 (1) > total Mabs: 3
From 2 RM: RM (Blood): gp140 (5), gp120 (1), gp41 (14), multi-Env (1) > total Mabs excluding multi-Env and gp140 (15)

It is important that the authors clarify in a table how many Env-specific Mabs were isolated in total from blood and milk of AGM, RM and humans as the percentages as plotted now give a skewed picture. Significantly more Env-specific Mabs need to be isolated from each animal and human to make a comparison and the results statistically significant and to make the genetic information shown in Figure 4 interesting. The low number of Mabs makes the overall conclusions drawn in this paper completely invalid.

9. There are a number of typos and grammar mistakes, thus the authors should proof-read the manuscript carefully.

Author's Response to Reviewers

Reviewer #1:

The manuscript by Zhang et al titled "the immunoglobulin repertoire of chronically simian immunodeficiency virus infected African green monkeys" is really quite an impressive body of work. The authors define the features of the AGM immunoglobulin loci and utilize this information to understand and compare the virus-specific memory B cell responses of AGMs to both humans and SIV-infected rhesus macaques. The authors denote some interesting differences in the proportions of SIV/HIV-specific responses in AGMs compared to SIV-infected RMs and humans, with AGMs having a higher proportion of Env-specific responses. The authors went further to try to understand the effector function of these Env-specific Abs in AGM and found that while they were poor at neutralizing virus they were robust at mediating ADCC and CD4-dependent virion capture. This review appreciates the enormous effort and detail that went into this work and only has minor comments and/or concerns listed below.

We thank the reviewer for his/her kind assessment of our manuscript. We agree that this study required an enormous amount of effort and detail, and we hope our study will provide a novel non-human primate antibody platform that will ultimately guide the development of gp120-directed vaccine responses by HIV vaccine concepts in humans.

1. While there appears to be significant differences in the proportional representation of Env-specific Ab responses in AGMs compared to RMs and humans, were these statistically significant? There appears to be an absence of formal statistical measures throughout the manuscript that either needs to be changed or explained.

We thank the reviewer for bringing this to our attention. In order to meet the character count limit from Nature Communications, we did not describe each statistical test throughout the manuscript. Instead, we included a summary of all statistical tests performed for each comparison for all figures in the supplemental material. For more details on the statistical analyses, please refer to table S8 at end of the supplementary materials. However, we also include the statistics on the comparison of antibody responses between AGMs, RMs and humans in Figure 3 and the figure legend. To provide a brief summary to the reviewer, the proportion of Env-reactive Abs in AGM and RM blood was 19.0% and 19.3%, respectively, and the difference between these proportions was not statistically significant. Interestingly, when we compared the proportion of Env-reactive Abs from the blood of AGMs (19%) to that of humans (0.5%), the difference was significant ($p < 0.0001$). Moreover, we report that 30.2% of AGM breast milk Abs were Env-reactive as compared to 7.3% Env-reactive Abs in humans, $p = 0.01$ (Fig. 3. and legend).

2. While the proportions of Env-specific Ab responses in AGMs was higher than RMs and humans was there a quantitative difference between the species? Did AGMs simply have a weaker response in which a higher proportion were Env-specific, or robust responses with higher proportions being Env-specific?

The reviewer raises an interesting question – whether AGMs have higher levels of Env-specific B cells simply because they have a higher total number of B cells or memory B cells compared to RMs. We compared the total number of CD3- CD20+ B cells at week 45, the closest timepoint available with complete blood count data to the timepoint of the B cell sort (week 52), between AGMs and RMs and saw no statistical difference ($p = 0.07$, Mann-Whitney test). Similarly, we compared the total number of memory B cells (as defined by CD19+ IgD-

CD27+) between AGMs and RMs and again saw no statistical difference ($p=0.25$, Mann-Whitney test). Thus, we conclude that AGMs have a higher proportion of Env-specific responses compared to RMs, and that this higher proportion of Env-specific responses is not due simply to a higher number in total B cells, or total memory B cells present in natural hosts as compared to RMs. We have included this analysis of the total number of memory and peripheral B cells between AGMs and RMs in the supplementary materials Table S10 and also in the results section lines 182-184.

3. The ADCC experiments certainly demonstrated increased effector function by this measure in AGMs on a per Ab level, however, was there any evidence in vivo for this robust effector function (i.e. activated NK cells, increased viral evolution or antibody evolution in the host, etc)?

We thank the reviewer for bringing up this critically important point. While we are highly interested in demonstrating the effect that these non-neutralizing AGM mAbs have on AGM viral evolution/immune pressure on the virus, we feel that this would be an entirely new research study in of itself. The purpose of our study was not to demonstrate that mAbs with non-neutralizing antibodies protect AGMs from disease progression. Instead, our goal was to describe the memory B cell populations in SIV-infected AGMs, generate a toolset/platform to generate and test AGM vaccine-elicited antibodies for their functional activity in vitro. To this end, we defined AGM VH+L gene usage of Env-specific mAbs and showed that these mAbs have functional anti-SIV activity in vitro. In addition, our findings in this study are concordant with previous studies from our group that showed that AGMs have distinct gp120/gp41 plasma kinetics as compared to SIV/HIV infected RMs and humans.

4. It is interesting the CD4-dependent virion capture was robust in AGMs. Does this mean that non-CD4 mediated virion capture was not? Do the authors believe this may reflect the reduced follicular dendritic cell SIV-immune complex trapping reported in AGMs? This might merit some discussion.

We thank the reviewer for bringing up this interesting point. Previous studies have reported that AGMs have reduced antibody-virion complexes that are trapped by follicular dendritic cells 1,2 (see references below). Given that antibody-virus complexes could expose conformational-dependent epitopes in the virus Env while it is engaged to antibodies, it is possible that this is a source of CD4i-dependent B cell responses. Given this hypothesis, it is possible that AGM mAbs that mediate virion capture in the presence of sCD4, but not in its absence, arise from B cells that were selected by follicular dendritic cells in the presence of immune-virus complexes. Conversely, it is possible that antibodies that do not mediate virion capture in the presence of sCD4 arise from B cells selected by follicular dendritic cells without immune-virion complexes. We have added a brief explanation of this hypothesis in our results in lines: 316-322.

Reviewer #2 :

Summary: Zhang et al describes the annotation of the immunoglobulin repertoire for African Green Monkeys. AGMs are of interest to HIV research because they are natural hosts for SIV-infection, but do not succumb to disease. The authors show differences in VDJ genes between AGMs, rhesus macaques (RMs), and humans. The authors then isolate monoclonal antibodies, which show stronger immunogenicity to gp120 than to gp41 for AGMs compared to humans or RMs. The HIV Ag-reactive monoclonal antibodies do not show any differences in levels of SHM or CDR3 lengths compared to the non-HIV Ag reactive antibodies. The monoclonal antibodies are also non-neutralizing, but do exhibit ADCC activity. General comments: The manuscript is somewhat cumbersome to read and the presentation of data does not fit a coherent message. The annotation of the AGM IgG is interesting and potentially useful. These data by themselves may not be sufficient to warrant publication in this journal, so it is understandable that subsequent experiments were performed to compare AGM responses to RM and humans. The results of these experiments,

however, are vague and the conclusions are over-interpreted. First, the rationale for AGMs as an alternative model for vaccine studies needs to be better articulated. If SIV-infection of AGMs does not lead to disease progression, why should they be used for vaccine studies? What evidence exists that disease control is antibody mediated (the authors actually list a series of references that would suggest that adaptive immunity is not important for disease control)? Highlighting differences in immune responses between humans and AGMs contradicts the use of these animals as vaccine models for expected responses in humans. There needs to be clearer goals for how or why the AGM antibody repertoire is of use to HIV vaccine design, otherwise the manuscript should be rewritten more simply as a description of the AGM antibody repertoire.

We thank the reviewer for requesting this clarification. Given their dominant gp120-directed antibody responses after SIV infection, we believe that AGMs could be championed as an NHP model to test HIV gp120-directed vaccine concepts. Particularly, the AGMs could inform how to direct vaccine-elicited responses to the gp120-portion of the HIV Env, as opposed to the high levels of gp41-specific responses that have been described for some HIV Env vaccine candidates. Interestingly, most broadly neutralizing epitopes have been mapped to gp120, yet a major roadblock in eliciting these responses is our lack of understanding of strategies on how to elicit them. Furthermore, recent landmark studies have demonstrated that HIV Env antibody responses during acute HIV infection and in human immunization studies that include gp140 in the vaccine insert predominantly target gp413 (see reference below). Importantly, these types of gp41-directed antibody responses are non-functional 3 4(see references below), thus strategies aimed at eliciting gp120-directed responses are critically needed. This issue is particularly important for trimer-based vaccine approaches which must include the gp41 domain. Interestingly, we have shown that chronically infected AGMs have a predominant gp120-directed response. We believe that more studies are needed to tease apart the mechanism of this unique gp120-directed bias in AGMs. We hope that elucidating these mechanisms will inform the development of HIV vaccines that elicit strong gp120-specific and minimal nonfunctional gp41-specific antibody responses.

While several characteristics specific to AGMs have been described as potential reasons why this species does not develop an AIDS like disease, it remains unclear if antibodies contribute to their lack of disease progression. Interestingly, our previous work shows that AGMs do not demonstrate the B cell dysfunction that is observed in HIV-infected humans and SIV-infected RMs. Regardless of the role of humoral immunity in disease progression in AGMs, a better understanding of the gp120-directed bias in this species could help guide development of HIV vaccine approaches that target gp120 epitopes. Second, what was the rationale for not antigen sorting memory B cells? Could there be neutralizing antibodies present if the repertoire was sampled more deeply?

We agree with the reviewer that if the Env-specific memory B cell repertoire was more deeply sampled using Env-specific hooks to sort single memory B cells, this may in fact isolate AGM SIV-specific neutralizing antibodies since we have previously demonstrated that autologous virus neutralization occurs within 6 weeks of SIV infection in these animals. However, for the purpose of our study investigating the unbiased memory B cell repertoire in natural SIV hosts, we investigated the proportion of Env-directed responses in AGMs by sampling the total memory B cell population. From the total memory B cell population, we cloned their VH and VL genes and screened their transiently transfected products against a panel of both HIV and SIV antigen panel as described in the methods. This approach will not bias the overrepresentation of gp120-specific or gp41-specific B cells that may result from antigen specific B cell sorting, but instead assesses the proportion of gp120 or gp41-specific B cells in total memory B cell population. This method, utilized across AGMs, RMs, and humans allowed comparison of the true proportion of memory B cells that are dedicated to SIV/HIV Env.

Third, is the general conclusion that non-neutralizing/ADCC-active antibodies in AGM might be responsible for disease control? If so, are the antibodies elicited in RM in the same study also non-neutralizing/ADCC-active? Would such a result in RM then negate this hypothesis for AGM?

The reviewer brings up an interesting point - does the presence of Env-specific non-neutralizing/ADCC mediating antibodies with antiviral activity contribute to lack of disease progression in AGMs compared to RMs? While this is a great scientific question, we cannot conclude this from our investigation of the antiviral activity of 10 Env-specific monoclonal antibodies that we functionally characterized from a limited number of SIV-infected AGMs. To more thoroughly address this question, it would be important to define the effect of AGM ADCC/non-neutralizing functional antibodies on driving virus evolution on autologous AGM SIV variants. We feel that delineating the degree of immune pressure that non-neutralizing antibodies in AGMs and RMs apply to autologous viruses is a separate study that is beyond the scope of this study. Our goal in this study is not to identify the role of AGM antibodies on the disease outcome in this species, but instead we intended to provide a view of the unbiased memory B cell populations in natural and non-natural hosts of SIV/HIV, as well as a novel NHP platform to investigate ways to elicit gp120-directed antibody responses. However, we discuss the possibility of the contribution of ADCC activity to the lack of disease progression in the discussion (lines 319-324) and we recognize the need for further studies regarding this very question.

Finally, it would be interesting/informative to perform competition assays between the monoclonal antibodies isolated from AGM to determine if they're binding the same epitope. Indeed, if AGMs are being championed as an alternative vaccine model, a better comparison of AGM vs RM repertoires and responses needs to be performed. If the argument is that these animal models have different antibody repertoires and that this is the reason for disease control, then immunization of AGMs and RM with the same gp120 protein antigen and the elicitation of different epitope responses would be more convincing to support this argument.

We thank the reviewer for this useful suggestion. To investigate if AGM gp120-specific ADCC-mediating mAbs with unknown fine-epitope specificity bind to the same epitope, we selected four gp120-specific AGM mAbs with undefined fine epitope specificity that mediated robust ADCC activity and performed a blocking/competition binding ELISA to SIVsabgp120. Interestingly, we found that DH546 and DH552 blocked binding to SIVsabgp120 against each other, but were not blocked by DH549 and DH550. Furthermore, DH549 and DH550 blocked binding to SIVsabgp120 against each other, but were not blocked by DH546 and DH550. The finding that these AGM mAbs pairs block binding of gp120 against each other, and that these antibodies mediate robust ADCC activity directed at gp120, suggests that these antibodies target similar or adjacent epitopes on gp120. We have added % blocking data for these AGM mAbs in the results (lines 251-255) and supplementary materials, table S9.

We agree that it would be interesting to assess if AGMs and RMs elicit distinct epitope-specific antibody responses when immunized with the same Env-immunogen. In fact, we have an ongoing study investigating this very question. Briefly, we are immunizing AGMs and RMs with a trimeric HIV gp140 envelope to determine if higher magnitude gp120-specific antibody responses are elicited in AGMs compared to RMs. Therefore, this is a separate line of investigation that has arisen as part of this work.

Specific comments: 1. Would be easier to comment on specific items if line numbers were used
We agree that line numbers should have been included in the first submission and we have added them.

2. In the abstract, "Here we define features of the AGM Ig loci and compare their virus-specific memory B cell repertoires to that of humans and RMs" - based on the figures and text, the memory B cells were not antigen sorted?

We see how reading this statement could mislead to reader to conclude that the memory B cells were antigen sorted. We have changed the text on the manuscript and removed "virus-specific". It should be noted that we did not perform antigen-specific B cell sorting in order to unbiasedly interrogate the proportion of gp120 vs gp41-specific memory B cells between AGMs, RMs, and humans. If antigen-specific B cell sorting was performed in our study, it is likely that the Env antigen used for sorting would result in a skewing of the isolated B cell populations to those responding to a particular conformation of Env. This unbiased memory B cell analysis that we performed in this work allowed comparison of the proportion of Env-specific B cell populations across natural and non-natural SIV/HIV hosts. Thus, we opted for investigating the proportion of gp120 and gp41-specific B cells from total memory B cells of each species.

3. The line, "Although more than half of chronically HIV infected individuals develop some degree of broadly neutralizing antibodies (bnAbs)" is inaccurate, the paper reports broadly neutralizing serum responses, not bnAbs

*We thank the reviewer for his correct interpretation of the cited study. We have updated this line in the manuscript to reflect that more than half of chronically HIV infected individuals develop broadly neutralizing **serum** responses.*

4. The lines, "Defining the roadmap for how epitope-specific bnAbs are produced will be important in designing potential strategies to induce broadly reactive HIV antibodies. Yet, defining this roadmap for eliciting gp120-directed bnAbs may require unique non-human primate models and B cell repertoire analysis tools." - how does the last sentence lead to the first sentence?

We agree with the reviewer that these two sentences together can be confusing. In response to the reviewer, we have changed the later sentence to "Here we introduce a unique non-human primate model and develop new tools to help define the elicitation of gp120-directed antibody responses, an initial step in eliciting gp120-directed neutralizing responses." We hope that this clarifies our point.

5. The line, "Together, these results suggest that SIV Env-reactive antibodies, particularly those that are gp120-specific, in AGMs may mediate in vivo antiviral function through non-neutralizing mechanisms that may be dependent on Env-CD4 engagement" is more speculation than a conclusion

We agree with the reviewer that we cannot make this conclusion from our data. We have changed the manuscript to a less speculative statement: "Together, these results suggest that AGM SIV Env-reactive antibodies can mediate robust non-neutralizing antiviral functions". Please refer to lines: 259-260 for an updated narrative.

6. The line, "We then utilized this platform to genetically characterize 175 mAbs isolated from the blood and milk memory B cells from four chronically SIV-infected AGMs." - is somewhat misleading please specify the numbers of mAbs that were Env-reactive (i.e. 44)

We agree with the reviewer that this could be a misleading statement; we have updated the manuscript to reflect the number of Env-reactive mAbs analyzed as opposed to the total number of AGM mAbs analyzed: "We then utilized this platform to genetically characterize 44 SIV-reactive mAbs isolated from the blood and milk memory B cells from four chronically SIV-infected AGMs." Please refer to line: 277.

7. The line, "Thus, interrogating the Env-specific memory B cell repertoire in AGMs may uncover mechanisms of this distinctly focused B cell response which may provide new clues for developing novel HIV vaccines that will direct the human B cell repertoire towards more desirable HIV Env epitope targets." - how specifically would evaluating the Env-specific B cell repertoire in AGM lead to this?

We believe that investigating the nature of the high magnitude gp120-specific B cells responses in the AGM primate model, in contrast to the undesirable gp41-specific response that is observed during acute infection and gp140 immunization in humans could provide clues for designing and developing novel HIV vaccines that are better able to direct responses against gp120 epitopes. Here we provide a novel platform to interrogate the evolution of this gp120-bias in SIV-specific antibody responses of AGMs to inform the field on how to induce gp120-directed responses.

Reviewer #3

(Remarks to the Author): The authors have identified and characterized the functional immunoglobulin repertoire in African Green Monkeys (AGM) using data from the recently published AGM genome. AGM is a natural host species for simian immunodeficiency virus (SIV). This manuscript clearly establishes that AGM Env-specific mBCs are predominantly reactive against gp120 in contrast to the more frequent gp41-directed responses of rhesus macaques and humans. These gp120-reactive mBCs in chronically SIV-infected AGMs exhibit surprisingly limited activity against non-autologous, variant SIV, yet are potent in ADCC and virus-capture activity against the autologous SIV challenge strain; as the authors suggest, a continued study of the role of these non-neutralizing antibody-mediated effector functions in a natural host could shed light on HIV clearance strategies in humans. The conclusions of this novel manuscript are appropriately discussed, and caveats included (e.g., the AGM genome is only a draft and is derived from a single monkey). Statistical treatment of the data (Wilcoxon tests and Fisher's Exact) are appropriate. Zhang et al. are experts in the field and have created a uniquely valuable data set which will allow for detailed immunogenetic investigations of SIV infection in AGM and potentially lead to new insights that may have bearing on gp120-directed HIV vaccine development.

We thank reviewer #3 for their kind assessment of our work. We also hope that our dataset will prove valuable for further detailed immunological investigations that may have implications in inducing gp120-specific antibody responses in human HIV vaccine studies.

Reviewer #4 :

The paper by Zhang et al. investigates the memory B cell compartment in chronically SIV-infected African green monkeys, which are natural hosts of (non-pathogenic) SIV infection. The approach taken is to isolate monoclonal antibodies (Mabs) from total IgD negative memory B cells and, following cloning and expression of antibody heavy and light chains, the antibody specificities were tested against various forms of Env. The authors conclude that "AGM Env-specific memory B cells were predominantly gp120-directed in contrast to the frequent gp41-directed responses of RMs and humans" and that co-evolution of SIV and AGMs has resulted in an SIV Env-specific B cell repertoire that differs from that observed in non-natural SIV/HIV infection. These are sweeping statements based on very little amount of data. Upon inspection of the results there are some major concerns that severely cripple the manuscript and make the conclusions drawn invalid. There are also several technical issues, as follows: 1. The title, "The immunoglobulin repertoire of chronically simian immunodeficiency virus infected African green monkeys", is misleading as the authors do not perform deep sequencing of large number of SIV-specific B cells. The study is focused on envelope-specific B cells and very few monoclonal antibodies are isolated and analyzed.

We agree with the reviewer that this title could be potentially confusing. While we did not deep sequence the VH and VL genes in antigen specific B cells in AGMs, we do describe Env-specific B cell population from the total memory B cell pool of SIV-infected AGMs. We have changed the title to "Env-specific memory B cell populations in chronically SIV-infected African Green Monkeys".

2. In Fig 2, the authors show a phylogenetic tree of rhesus macaque VH genes consisting of 61 different genes. However, the authors refer to a paper by Francica et al, which describes over 100 rhesus macaque V genes. Thus, there seems to be a mismatch in the numbers between what is referenced and what is shown, why is this? Which of these are actual genes and which are allelic variants of the same gene? How many animals were sequenced to define the database published by Francica et al? This should be clarified.

*We agree with the reviewer that there should be more clarity on which rhesus V gene database is being compared to the AGM V genes. While the recent study published by Francica et al., describes over 100 rhesus macaque VH genes, it should be noted that this number includes allelic variants of VH genes. The true number of VH genes that was reported by Francisca et al. is 58. The extra 42 VH genes represent allelic variants of the 58 VH genes in the Francica et al., study. We agree that failing to make this distinction leads to a falsely perceived notion that we have not defined a correct number of VH genes. When we defined VH genes, only the first allele (*01) from human and rhesus VH genes was used to make the trees reported in our manuscript. Thus, the additional 42 VH alleles were not included in our study. We realized how this could have led to confusion, as we did not clarify this in our methods section. Additionally, the Francica et al., study sequenced 10 rhesus macaques likely adding to their identification of additional allelic variants. Thus, we updated our analysis to include this database of rhesus VH genes and we have discussed this in an updated methods section (Figure: 2). We updated this number to 58 rhesus VH genes in Figure 2.*

3. The authors compile a database for AGM VH genes based on reference 13. How many animals were analyzed in ref 13? Were the germline versions of all the V genes in the AGM database shown to be expressed in naïve B cells of the AGMs used in the current study? Did the authors look at this?

Only 1 AGM genome was analyzed in ref 13, and we point out that using one animal to define the immunoglobulin locus is a caveat to our study (lines 271-276). Yet, based on the available annotated genome sequencing, we have developed new tools to interrogate monoclonal antibody responses in AGMs. Additionally, we identified variable germline genes by identifying the rearrangement signal sequence (RSS) in the AGM genome. We defined functional V genes based on the human functional RSS sequences that we confirmed to be functional using the human IgG gene database. See lines: 384-392. Therefore, our immunoglobulin gene annotation will serve as a basis for further updates as additional AGM genomes are sequenced and annotated.

4. The way the percentages in Figure 3 are calculated is confusing in that percentages of antibodies that bind gp140 are added to the percentages of antibodies that bind gp120 and gp41. These specificities should be largely overlapping and they are therefore "counted twice". This should be revised and the data replotted.

We appreciate that our notation could be confusing, but no Env-specific antibodies were "counted twice" in our analysis or description. The gp140 antibodies were the antibodies bound to gp140 only, but not gp120 or gp41. We have listed the definition of the different antibody groups in the updated Fig 3. Recent studies have reported that Env-specific antibodies bind to the interface of gp140 5 (see reference below). Specifically, interface-directed antibodies bind to gp140, but do not bind to gp120 or gp41 alone, suggesting that they bind to interface of gp140. While, we cannot conclusive state that the AGM antibodies that bind to gp140, but not gp120 or gp41 are definitely "interface"

antibodies, it is possible that they are. We did not count the antibodies twice, and we have updated the Fig 3. with the definition of different antibody classifications in the updated version to make that more clear.

5. The fraction referred to as multi-Env is also confusing. According to the figure legend, this indicates antibodies that bind both gp41 and gp120, which does not make sense. This fraction is also added to the other specificities so again, specificities are counted more than once. Please clarify the intention behind this.

*We thank the reviewer for asking for this clarification. Again, we did not count or report any group of the Env-specific antibodies more than once. We agree that the multi-Env term can be confusing and have renamed antibodies that bound to gp120 and gp41 to **Env-cross reactive** antibodies. When this class of antibodies was screened against gp140, gp120 and gp41, the majority bound to gp140, gp120 and gp41 above our strict background cut off (see Methods lines 433-434). Interestingly, seven antibodies Env cross-reactive mAbs did not bind to gp140, yet bound to both gp120 and gp41. We now refer to these antibodies as Env cross-reactive in the manuscript. Please see the updated Fig. 3 with the Env-cross reactive antibodies and a corresponding table showing the tabulation of the frequency of these antibodies. Please refer to Table S7 for genetic information and binding data of AGM antibodies described in this manuscript.*

6. In the description of the ELISAs, it says that gp41 reactivity of some Mabs was measured against a linear peptide, rather than against folded protein. This is not ideal as all gp41 reactivities would not be detected this way. This should be commented on.

We thank the reviewer for pointing out this important limitation in the screening gp41 reactive specificities, as few conformational gp41 reagents exist. While we used a linear gp41 peptide previously shown to be immunodominant in AGMs 6,7 (see references below) in the antibody screening ELISA for AGMs, and RMs B cell transient transfection supernatants were screened with the same homologous SIV/HIV linear gp41 peptide. Thus, it is unlikely that using this peptide confounds our interpretation that AGMs mount a gp120-focused antibody response, compared to the gp41-focused response in RMs and humans.

7. In the ELISAs used to determine the specificities of anti-HIV-1 Mabs, consensus Env proteins were used. While this may be the best one can do given that these individuals harbor diverse virus populations, Cons probes are not going to pick up all reactivities. This may lead to an underestimation of the frequencies of Env-specific memory B cells, which should also be commented on.

We agree with the reviewer that using consensus HIV Env probes in the screening ELISAs could lead to an underestimation of the frequencies of Env-specific memory B cells in humans, and this is discussed as a caveat in lines 295-300. We used the autologous gp140, gp120 and gp41 envelope proteins to screen RM and AGM heavy and light chain pairs. Yet for humans, given the wide viral diversity within an individual, and the cost involved in generating a large number of autologous antigens in HIV infected individuals, the HIV Cons probe was the best consensus antigen available to us.

8. Importantly, because total memory B cells were sorted rather than using specific probes to pull out Env-specific memory B cells, the number of specific Mabs the authors were able to isolate and analyze is very low. Especially if some of the MABs are counted more than once. Removing the Multi-Env MABs that also bind other probes the data in Figure 4 demonstrate that only 12 Mabs were isolated from AGM blood and 15 from AGM milk, while from humans only 1 Mab was isolated from

blood and 3 from milk. Thus, from five humans included in the study, only a total of 3 Env-specific antibodies were isolated. These are far too low numbers to base any conclusions on. Based on Figure 4, the total number of Mabs appears to be: From 4 AGM: AGM (Blood): gp120 (9), gp41 (3), multi-Env (3)> total Mabs: 15 (12 if excluding multi-Env); AGM (Milk): gp120 (10), gp41 (5), multi-Env (14)> total Mabs: 29 (15 if excluding multi-Env) From 5 humans: Human (Blood): gp41 (1); Human (Milk): gp120 (2), gp41 (1)> total Mabs: 3 From 2 RM: RM (Blood): gp140 (5), gp120 (1), gp41 (14), multi-Env (1) > total Mabs excluding multi-Env and gp140 (15) It is important that the authors clarify in a table how many Env-specific Mabs were isolated in total from blood and milk of AGM, RM and humans as the percentages as plotted now give a skewed picture. Significantly more Env-specific Mabs need to be isolated from each animal and human to make a comparison and the results statistically significant and to make the genetic information shown in Figure 4 interesting. The low number of Mabs makes the overall conclusions drawn in this paper completely invalid.

We thank the reviewer for bringing this lack of clarity to our attention. We have a summary table listed in Figure 3 with all the antibodies enumerated and categorized and we have also added the strict and non-overlapping definition of each antibody groups. We hope the renaming of the Multi Env category as Env cross-reactive and further clarification that no Env-specific mAbs were “counted twice” decreases the concern of over-representing Env-specific mAbs and lack of clarity.

9. There are a number of typos and grammer mistakes, thus the authors should proof-read the manuscript carefully.

We thank the reviewer for pointing this out. We have assiduously checked the entire manuscript and supplementary materials and corrected all typos and grammatical errors.

1 Brenchley, J. M. *et al.* Differential infection patterns of CD4+ T cells and lymphoid tissue viral burden distinguish progressive and nonprogressive lentiviral infections. *Blood* **120**, 4172-4181, doi:10.1182/blood-2012-06-437608 (2012).

2 Beer, B. *et al.* Lack of dichotomy between virus load of peripheral blood and lymph nodes during long-term simian immunodeficiency virus infection of African green monkeys. *Virology* **219**, 367-375 (1996).

3 Williams, W. B. *et al.* HIV-1 VACCINES. Diversion of HIV-1 vaccine-induced immunity by gp41-microbiota cross-reactive antibodies. *Science* **349**, aab1253, doi:10.1126/science.aab1253 (2015).

4 Trama, A. M. *et al.* HIV-1 envelope gp41 antibodies can originate from terminal ileum B cells that share cross-reactivity with commensal bacteria. *Cell Host Microbe* **16**, 215-226, doi:10.1016/j.chom.2014.07.003 (2014).

5 Huang, J. *et al.* Broad and potent HIV-1 neutralization by a human antibody that binds the gp41-gp120 interface. *Nature* **515**, 138-142, doi:10.1038/nature13601 (2014).

6 Amos, J. D. *et al.* Rapid Development of gp120-Focused Neutralizing B Cell Responses during Acute Simian Immunodeficiency Virus Infection of African Green Monkeys. *J Virol* **89**, 9485-9498, doi:10.1128/JVI.01564-15 (2015).

7 Amos, J. D. *et al.* Lack of B cell dysfunction is associated with functional, gp120-dominant antibody responses in breast milk of simian immunodeficiency virus-infected African green monkeys. *J Virol* **87**, 11121-11134, doi:10.1128/JVI.01887-13 (2013).

Reviewers' comments

Reviewer #1 (Remarks to the Author):

I believe that the authors have appropriately addressed all of the reviewers comments and concerns, which has strengthened the manuscript.

Reviewer #2 (Remarks to the Author):

The authors have responded to earlier comments.

Reviewer #3 (Remarks to the Author):

This referee has carefully read the point-by-point response letter; I am satisfied that the points raised in the previous round of review have been satisfactorily addressed by the authors. The authors had already explicitly addressed caveats to their study in the original submission, and any calls for additional experimentation (Referee #2) would in my opinion be far beyond the scope of a communication.

Akin to Referee #1, I had only minor criticisms of what is otherwise an impressive data-set and substantial amount of work presented in this excellent manuscript. I am confident that the field of HIV vaccine design will cite this study and that - like the best of research - it will pave the way for new, important inquiry.

Reviewer #4 (Remarks to the Author):

Zhang et al. have revised and improved their manuscript in response to some of the concerns raised by the reviewers. The title is changed and the terminology used for the different sub-specificities within the Env-specific responses determined by ELISA has been clarified. Furthermore, a table has been added demonstrating how many MABs were isolated. This helps, however, because total memory B cells were sorted rather than Env-specific memory B cells, the number of specific MABs the authors were able to study is very low (the total number of human MABs listed in Fig 3c is only 1 for human blood and 3 for human milk), which is still a major weakness of the paper. This makes the conclusion "AGM Env-specific memory B cells were predominantly gp120-directed, in contrast to the frequent gp41-directed responses of RMs and humans" a major concern. If this paper were to be published, the authors should tone down this statement as this comparison is not possible to make based on such a limited number of MABs.

Author Response to Reviewers

Reviewer #4:

Zhang et al. have revised and improved their manuscript in response to some of the concerns raised by the reviewers. The title is changed and the terminology used for the different sub-specificities within the Env-specific responses determined by ELISA has been clarified. Furthermore, a table has been added demonstrating how many MABs were isolated. This helps, however, because total memory B cells were sorted rather than Env-specific memory B cells, the number of specific MABs

the authors were able to study is very low (the total number of human MAbs listed in Fig 3c is only 1 for human blood and 3 for human milk), which is still a major weakness of the paper. This makes the conclusion "AGM Env-specific memory B cells were predominantly gp120-directed, in contrast to the frequent gp41-directed responses of RMs and humans" a major concern. If this paper were to be published, the authors should tone down this statement, as this comparison is not possible to make based on such a limited number of MAbs.

We thank reviewer #4 for his insightful comment. We agree that we should tone down these types of statements directly comparing the frequency of gp120 and gp41-responses between AGMs and RMs/humans. As an example, we have changed the conclusion noted above to the following: 'We demonstrate that AGMs have a higher proportion of Env-specific memory B cells that are mainly gp120-directed'. We have also 'toned down' other similar statements. See lines: 93-95. "Our findings revealed that differences in the frequency of gp120-directed antibody responses in AGMs compared to RMs and humans" to the following: Our findings reveal that AGMs have a higher proportion of Env-specific memory B cells.